 

# CYK-4 functions independently of its centralspindlin partner ZEN-4 to cellularize oocytes in germline syncytia

Kian-Yong Lee[1†], Rebecca A Green[1†], Edgar Gutierrez[2], J Sebastian Gomez-Cavazos[1], Irina Kolotuev[3‡], Shaohe Wang[1§], Arshad Desai[1], Alex Groisman[2], Karen Oegema[1*]

[1]Ludwig Institute for Cancer Research, Department of Cellular and Molecular Medicine, University of California, San Diego, La Jolla, United States; [2]Department of Physics, University of California, San Diego, La Jolla, United States; [3]Microscopy Rennes Imaging Center and Biosit, University of Rennes 1, Rennes, France

**\*For correspondence:**
koegema@ucsd.edu

[†]These authors contributed equally to this work

**Present address:** [‡]Facultéde biologie et de médecine, Electron Microscopy Facility, University of Lausanne, Lausanne, Switzerland; [§]Cell Biology Section, National Institute of Dental and Craniofacial Research, National Institutes of Health, Bethesda, United States

**Competing interests:** The authors declare that no competing interests exist.

**Abstract** Throughout metazoans, germ cells undergo incomplete cytokinesis to form syncytia connected by intercellular bridges. Gamete formation ultimately requires bridge closure, yet how bridges are reactivated to close is not known. The most conserved bridge component is centralspindlin, a complex of the Rho family GTPase-activating protein (GAP) CYK-4/MgcRacGAP and the microtubule motor ZEN-4/kinesin-6. Here, we show that oocyte production by the syncytial *Caenorhabditis elegans* germline requires CYK-4 but not ZEN-4, which contrasts with cytokinesis, where both are essential. Longitudinal imaging after conditional inactivation revealed that CYK-4 activity is important for oocyte cellularization, but not for the cytokinesis-like events that generate syncytial compartments. CYK-4's lipid-binding C1 domain and the GTPase-binding interface of its GAP domain were both required to target CYK-4 to intercellular bridges and to cellularize oocytes. These results suggest that the conserved C1-GAP region of CYK-4 constitutes a targeting module required for closure of intercellular bridges in germline syncytia.

DOI: https://doi.org/10.7554/eLife.36919.001

## Introduction

During gametogenesis in the male and female germlines of metazoan species, dividing cells often undergo incomplete cytokinesis to form clusters of cells connected by stable intercellular bridges (*Haglund et al., 2011*). In *Drosophila*, where pioneering work on syncytial architecture has been done, germline intercellular bridges are called ring canals (*Robinson and Cooley, 1997*). The conservation across diverse metazoan species and essential role in fertility (*Greenbaum et al., 2006*; *Haglund et al., 2011*; *Lei and Spradling, 2016*; *Robinson and Cooley, 1997*) suggest that syncytial germline architecture, in which multiple nuclei reside in compartments that share a common cytoplasm, is important for the generation of viable gametes. The benefits of communal living for germ cell nuclei is an active area of investigation. However, one role for cellular interconnectivity, demonstrated for the female germlines in *Drosophila* and mice, is to allow the nuclei in adjacent connected cells to serve a nurse function in which they assist in the generation of the large quantities of mRNA, protein, and organelles that are required to properly provision developing oocytes (*King and Mills, 1962*; *Lei and Spradling, 2016*; *Pepling, 2016*; *Robinson and Cooley, 1997*).

The stable intercellular bridges in syncytial structures are generated by incomplete cytokinesis and contain some of the components found at intercellular bridges that are resolved by abscission during normal cytokinesis (*Haglund et al., 2011*). In dividing somatic cells, the contractile ring constricts down around the central spindle, a structure composed of a set of anti-parallel microtubule

bundles that forms between segregating chromosomes and concentrates key molecules that promote contractile ring assembly. As constriction completes, the central spindle is converted into a microtubule-based midbody, and the contractile ring is converted into a midbody ring. During normal cell division, the midbody and midbody ring work together to promote abscission, which cuts the intercellular bridge to form the two daughter cells (*Agromayor and Martin-Serrano, 2013*; *Mierzwa and Gerlich, 2014*). During the formation of germline cysts, the contractile ring constricts down around the central spindle, which forms a midbody, but abscission does not occur. Instead, the midbody is disassembled leaving a stable open intercellular bridge (*Dym and Fawcett, 1971*; *Weber and Russell, 1987*).

Ultrastructurally, all intercellular bridges contain an electron dense layer immediately beneath the plasma membrane (*Haglund et al., 2011*). In the female, but not the male, germline in *Drosophila*, ring canals undergo a maturation process in which they increase in size, recruit additional components and assemble an inner, less electron dense, layer composed of bundled circumferentially organized actin filaments. Presumably, this actin layer reinforces the canals to facilitate the transfer of cytoplasm and organelles into the developing oocyte (*Robinson and Cooley, 1997*). As a similar inner layer has not been observed in other systems, it is not yet clear if all intercellular bridges employ this mechanism for structural reinforcement. The intercellular bridges connected to the oocyte must eventually close to cellularize the egg and separate it from the germline syncytium. How intercellular bridges/ring canals are reactivated to close in the absence of an intervening central spindle is an important unanswered question.

The most conserved component of intercellular bridges, observed in all bridges examined, is the centralspindlin complex (*Carmena et al., 1998*; *Greenbaum et al., 2009*, *2007*, *2006*; *Haglund et al., 2010*; *Minestrini et al., 2002*; *Zhou et al., 2013*). The contractile ring components anillin and the septins are also often, but not always, present in these structures (*Haglund et al., 2011*). Centralspindlin is a heterotetrameric complex composed of a dimer of kinesin-6 (MKLP1 in humans, Pavarotti in *Drosophila*, ZEN-4 in *Caenorhabditis elegans*), and a dimer of a Rho family GTPase-activating protein (CYK4/MgcRacGAP/RACGAP1 in humans, CYK-4 in *C. elegans*, and RacGAP50C/Tum in *Drosophila*) (*White and Glotzer, 2012*). In *C. elegans*, the N-terminal half of CYK-4 has a coiled-coil that mediates its dimerization, and a short region prior to the coiled-coil that mediates association of the CYK-4 dimer with ZEN-4 (*Figure 1A*; (*Mishima et al., 2002*; *Pavicic-Kaltenbrunner et al., 2007*). The C-terminal half of CYK-4 contains its GAP domain and an adjacent C1 domain that is expected, based on work on human Cyk4 (*Lekomtsev et al., 2012*), to bind polyanionic phosphoinositide lipids and mediate association with the plasma membrane. ZEN-4 has an N-terminal motor domain and a C-terminal coiled-coil-containing region that mediates its dimerization and association of the ZEN-4 dimer with CYK-4 (*Figure 1A*; (*Mishima et al., 2002*; *Pavicic-Kaltenbrunner et al., 2007*). Early in cytokinesis, centralspindlin localizes to and is required for the formation of the central spindle; from this location, it is also thought to play a role in the local activation of the small GTPase RhoA to promote contractile ring assembly (*Green et al., 2012*; *White and Glotzer, 2012*). Consistent with a role in activating RhoA, a region in the N-terminal half of human Cyk4, between the coiled-coil and C1 domains, has been shown to be phosphorylated by Plk1 to promote its binding to the RhoA GEF Ect2 (*Burkard et al., 2009*; *Wolfe et al., 2009*). Later in cytokinesis, the central spindle matures to form the microtubule-based midbody. In *C. elegans* embryos, examination of the intercellular bridge by 3D electron tomography at specific timepoints following the onset of furrowing has revealed that midbody microtubules are lost ~300 s prior to abscission (*König et al., 2017*). Despite the absence of midbody microtubules, ZEN-4 levels in the intercellular bridge increase prior to abscission (*Green et al., 2013*). In embryos in which assembly of the central spindle is prevented, ZEN-4 is not present on midzone/midbody microtubules during cytokinesis but is still recruited to the intercellular bridge prior to abscission (*Green et al., 2013*). These results suggest that centralspindlin can target to the intercellular bridge independently of midbody microtubules during abscission in *C. elegans*, perhaps consistent with the localization of centralspindlin to stable intercellular bridges that lack intervening midbodies in the syncytial germlines of different species (*Carmena et al., 1998*; *Greenbaum et al., 2009*, *2007*, *2006*; *Haglund et al., 2010*; *Minestrini et al., 2002*; *Zhou et al., 2013*).

Here, we analyze the role of centralspindlin at intercellular bridges in the *C. elegans* syncytial oogenic germline. During normal cytokinesis, both centralspindlin subunits have an equivalent essential role in assembly of the central spindle and contractile ring constriction (*Canman et al., 2008*;

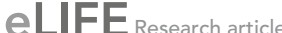

**Figure 1.** Both centralspindlin subunits localize to intercellular bridges throughout *C. elegans* germline development. (**A**) (*Left*) Schematics highlight the domain structure of the two molecular components, CYK-4 and ZEN-4, of the heterotetrameric centralspindlin complex, and the location of the temperature sensitive mutations used in this study. (*Right*) Schematic showing that centralspindlin is formed by the association of a dimer of CYK-4 with a dimer of ZEN-4. (**B**) (*Left*) Schematics illustrate the development of the syncytial germline. See text for details. (*Right*) Fluorescence confocal images of

*Figure 1 continued on next page*

*Figure 1 continued*

germlines in worms at the indicated stages expressing an mCherry-tagged plasma membrane marker and either CYK-4::mNeonGreen or *in situ*-tagged GFP::ZEN-4. Images for L4 and adult stage germlines are maximum intensity projections. (C) Rachis bridges increase in diameter during oocyte loading prior to their closure during cellularization. (*Upper left*) Schematic shows the location of the germline regions in the images and the nomenclature for labeling the compartments in the Loop (Loop 1–4) and Oocyte Cellularization (Oocyte 1–5) Regions. As indicated, Oocyte 1 was the first compartment after the turn. (*Upper right*) Graph plotting the diameters of the rachis bridges (mean ± SEM) in the indicated regions of the germline measured in the CYK-4::mNeonGreen images. n = number of cells analyzed at the indicated position/region. (*Lower panels*) Maximum intensity projections of confocal images of the pachytene, loop, and oocyte cellularization regions of adult germlines acquired in the strain expressing an mCherry-tagged plasma membrane marker and CYK-4::mNeonGreen. Merged images are shown alongside single color images showing the CYK-4::mNeonGreen signal. Scale bars are 10 μm.

DOI: https://doi.org/10.7554/eLife.36919.002

The following source data and figure supplements are available for figure 1:

**Source data 1.** Rachis bridges increase in diameter during oocyte loading prior to their closure during oocyte cellularization.

DOI: https://doi.org/10.7554/eLife.36919.005

**Figure supplement 1.** Generation of a functional single-copy transgene encoding CYK-4::mNeonGreen and *in situ*-tagged GFP::ZEN-4.

DOI: https://doi.org/10.7554/eLife.36919.003

**Figure supplement 1—source data 1.** The RNAi-resistant transgene encoding CYK-4::mNeonGreen rescues depletion of endogenous CYK-4.

DOI: https://doi.org/10.7554/eLife.36919.006

**Figure supplement 2.** Structure of the nascent syncytial germline and rachis in the L1 larva.

DOI: https://doi.org/10.7554/eLife.36919.004

*Davies et al., 2014*; *Jantsch-Plunger et al., 2000*; *Lewellyn et al., 2011*; *Loria et al., 2012*; *Mishima et al., 2002*; *Pavicic-Kaltenbrunner et al., 2007*; *Powers et al., 1998*; *Raich et al., 1998*; *Severson et al., 2000*). In contrast, we show that during the cytokinesis-like event in the germline that closes intercellular bridges to celluarize oocytes, CYK-4 is essential, whereas the kinesin-6 ZEN-4 is not. We show that the Rho GTPase binding interface of the CYK-4 GAP domain and its lipid-binding C1 domain are individually required to target CYK-4 to the rachis bridges and for oocyte cellularization. Consistent with the idea that recruitment to bridges is mediated by binding of the CYK-4 GAP domain to RhoA, knockdown of Rac and Cdc42 did not affect CYK-4 recruitment or oocyte cellularization, whereas RhoA inhibition disrupted germline structure in a fashion similar to CYK-4 inhibition. Cumulatively, our results highlight a critical role for CYK-4 in the closure of intercellular bridges during oocyte cellularization and suggest the C-terminal C1 and GAP domains of CYK-4 form a module required for targeting CYK-4 to cortical structures.

## Results

### Both centralspindlin subunits localize to intercellular bridges throughout *C. elegans* germline development

In *Drosophila* and mice, cells in syncytial germline cysts are connected to each other by stable intercellular bridges. One (*Drosophila*) or several (mice) of the most-connected cells in the cyst differentiate to become oocytes and the other cells serve a nurse-like function ( *Lei and Spradling, 2016*; *Robinson and Cooley, 1997*). The oogenic germline in the adult *C. elegans* hermaphrodite operates in a more egalitarian, assembly line-like fashion with ~1000 nuclei in cup-shaped cellular compartments connected to a common cytoplasmic channel called the rachis (*Hall et al., 1999*; *Hirsh et al., 1976*; *Lints and Hall, 2009*). Mitotic divisions at the distal tip generate new nuclei-containing compartments that progress through the stages of meiotic prophase as they traverse through the pachytene, loop, and oocyte cellularization regions (*Figure 1B,C*). Transcription in pachytene nuclei produces material that is loaded by cytoskeleton-driven flow into expanding compartments in the loop region (*Gibert et al., 1984*; *Starck and Brun, 1977*; *Wolke et al., 2007*). In the oocyte cellularization region, the circular openings into the nuclear compartments (which we will refer to as rachis bridges) close to separate individual oocytes from the syncytium, with one oocyte cellularizing approximately every 20 min. Consistent with prior work (*Zhou et al., 2013*), imaging of a functional CYK-4::mNeonGreen fusion and an *in situ*-tagged GFP::ZEN-4 fusion (*Figure 1—figure supplement 1*) revealed that both proteins localize to the surface of the rachis and to the rachis bridges (*Figure 1B,C*); both proteins are also detected at lower levels in nuclei. Consistent with prior

measurements (*Rehain-Bell et al., 2017*), rachis bridge diameter decreased by ~40% during progression through pachytene (from ~2.5 in the distal region to 1.4 µm in the proximal region) and then steadily increased concurrent with oocyte loading in the loop region (*Figure 1C*, Loop 1 to Loop 4). In the oocyte cellularization region, rachis bridge diameter decreased from ~4 µm to closed coincident with tapering of the rachis to a fine tip (*Figure 1C*, Oocyte 1 to Oocyte 4).

Monitoring of the two centralspindlin components during development of the germline syncytia (*Figure 1B*, Germline development (Larval Stages)) suggested that the cytoplasm-filled rachis is an expanded intercellular bridge. The germline arises from a pair of connected primordial germ cells ($Z_2$ and $Z_3$) that arise from an incomplete cytokinesis in the embryo (*Goupil et al., 2017*; *Hubbard and Greenstein, 2005*) and remain quiescent until the mid-L1 larval stage. At the L1 larval stage, the intercellular bridge connecting $Z_2$ and $Z_3$ extends out on one side (*Figure 1B*, *Figure 1—figure supplement 2*). Examination of serial sections of an L1 germline using electron microscopy revealed that the two nuclei-containing compartments sit side-by-side and open into the small cytoplasm containing intercellular bridge (the nascent rachis; *Figure 1—figure supplement 2*; *Video 1*). Subsequent nuclear divisions at the L1 and L2 stages generate additional compartments each with a circular opening to the rachis. At the L3 stage, the germline narrows in the middle, partitioning the syncytium into two arms that increase in length and fold back on themselves during the L4 stage. A dramatic structural reorganization of the germline accompanies the transition to oocyte production in the adult; the rachis increases in width to support component delivery to the nascent oocytes (*Figure 1B*). Thus, there is a structural continuity through development between the stable intercellular bridge arising from the first incomplete cytokinesis and the rachis in the adult. Consistent with the idea that the rachis is an extended intercellular bridge, the two centralspindlin subunits, which are conserved components of intercellular bridges throughout metazoans, localize to both the rachis surface and rachis bridges throughout development of the germline syncytia (*Figure 1B*).

## CYK-4 is required for oocyte cellularization in the adult germline

There are two distinct cytokinesis-like processes that contribute to germline development: (1) compartment proliferation: incomplete cytokinesis-like events that generate compartments, which remain attached via open bridges to the rachis, during development and at the distal tip in adults and (2) oocyte cellularization: complete cytokinesis-like events that occur in the proximal region of the adult germline, where rachis bridges close to cellularize oocytes and separate them from the germline syncytium. CYK-4 depletion has previously been shown to disrupt germline structure and cause sterility (*Green et al., 2011*; *Zhou et al., 2013*). To determine if *cyk-4* inhibition compromises either (or both) of the cytokinesis-like processes in the germline, we began by using a temperature-sensitive mutant in the C-terminal half of CYK-4 (*or749ts*; *Figure 1A*) that compromises both the GAP and C1 domains at the non-permissive temperature (*Canman et al., 2008*; *Davies et al., 2014*; *Zhang and Glotzer, 2015*). The product of *cyk-4(or749ts)*, CYK-4[E448K], interacts with ZEN-4 and supports assembly of the central spindle; nonetheless, upshift of *cyk-4 (or749ts)* embryos leads to a cytokinesis defect comparable in severity to CYK-4 depletion (*Canman et al., 2008*; *Davies et al., 2014*; *Loria et al., 2012*; *Zhang and Glotzer, 2015*). Prior work has also shown that a GFP fusion with CYK-4[E448K] fails to target to the rachis surface in the germline at the non-permissive temperature (*Zhang and Glotzer, 2015*), predicting that this mutant would exhibit a centralspindlin-null phenotype in the germline.

To assess the effect of CYK-4 C-terminal mutant on compartment proliferation during development, we upshifted L1 larvae to the non-

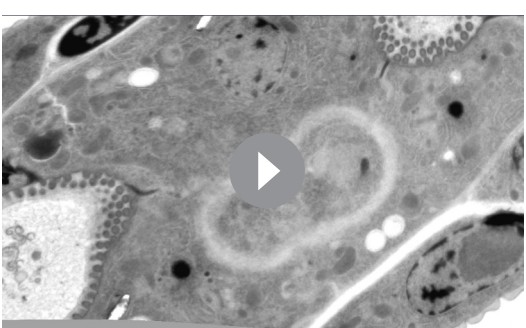

**Video 1.** Structure of the syncytial germline at the two-compartment stage in an L1 larva. Video shows images of a stack of serial 100 nm serial sections of an L1 germline that were collected and imaged by transmission electron microscopy. Images are shown without and then with superposition of a pseudocolor model in which the regions within the boundary of the two-compartment germline syncytium (*red*) and germline nuclei (*green*) are highlighted.
DOI: https://doi.org/10.7554/eLife.36919.007

permissive temperature (25°C), when the germline consists of a single pair of connected cells, and examined them at the L4 stage. Surprisingly, despite the penetrant effect of cyk-4(or749ts) on cortical remodeling during embryonic cytokinesis, germlines in L1-upshifted animals appeared normal at the L4 stage, with a comparable number of compartments to the germlines in control animals (*Figure 2A*). Consistent with the normal appearance of the germlines in the upshifted animals, L1 larvae that developed to the L4 stage at 25°C and were then shifted back to the permissive temperature (16°C) laid a normal number of embryos (*Figure 2A*).

In the converse experiment, performed to assess the effect of CYK-4 C-terminal mutant in the adult, larvae were grown to the L4 stage at the permissive temperature and were then shifted to the non-permissive temperature and analyzed 28 hr later. Under this regime, germline structure was strongly perturbed compared to equivalently treated control worms. Partitions were absent in the loop and oocyte cellularization regions and the proximal region of the germline had the appearance of a hollow multinucleated tube (tubulated germline; *Figure 2B*). These results indicate that the CYK-4 C-terminal region, which is required for the recruitment of CYK-4 to the rachis surface/ bridges, is important for germline morphology and function after the transition to oocyte production in the adult but is not required for compartment proliferation between the L1 and L4 larval stages.

## ZEN-4 is not essential for oocyte production in the adult syncytial germline

The above data show that a mutation in the CYK-4 C-terminus significantly disrupts embryo production by the adult syncytial germline. To determine if this function requires CYK-4 to act in the context of the centralspindlin complex, we depleted CYK-4 or ZEN-4 by injecting dsRNA at the L4 stage and analyzed germline structure 48 hr later. Consistent with the phenotype observed for the *cyk-4 (or749ts)* mutant shifted to the non-permissive temperature at the L4 stage (*Figure 2B*), partitions in the loop and oocyte cellularization regions were absent in *cyk-4(RNAi)* worms and the proximal region of the germline had the appearance of a hollow multinucleated tube (*Figure 3A*). As expected for such a severe defect, *cyk-4(RNAi)* worms were unable to produce embryos (*Figure 3B*). However, in striking contrast to CYK-4 depletion, ZEN-4 depletion had very little effect on germline structure (*Figure 3A,C*) or embryo production (*Figure 3B*). We confirmed using *in situ*-tagged GFP:: ZEN-4 that the RNAi conditions employed efficiently depleted ZEN-4 from the germline (*Figure 3C*); in addition, *zen-4* RNAi led to 100% embryonic lethality (*Figure 3B*), as expected based on its essential role in embryonic cytokinesis (*Mishima et al., 2002*; *Powers et al., 1998*; *Raich et al., 1998*). These observations suggest that CYK-4 can function in the adult germline independently of ZEN-4. As this result would predict, mNeonGreen-tagged CYK-4 maintained its localization on the rachis surface and rachis bridges in ZEN-4 depleted worms (*Figure 3D*), consistent with a prior immunofluorescence analysis showing that CYK-4 maintains its localization to the rachis surface in an upshifted temperature-sensitive *zen-4* mutant (*Zhou et al., 2013*).

The above result suggests that, in contrast to the *cyk-4(or749ts)* mutant that disrupts the CYK-4 C-terminus, the fast-acting temperature-sensitive mutants that block centralspindlin assembly by disrupting the interaction between the CYK-4 and ZEN-4 dimers (*cyk-4*(*t1689ts*) and *zen-4*(*or153ts*), *Figure 1A*; [*Encalada et al., 2000*; *Gönczy et al., 1999*; *Jantsch-Plunger et al., 2000*; *Severson et al., 2000*]) might not disrupt germline structure in the adult. Consistent with this prediction, germline structure appeared normal in adult *cyk-4*(*t1689ts*) and *zen-4*(*or153ts*) mutant worms upshifted for 28 hr beginning at the L4 stage, and the upshifted mutant worms retained the ability to lay embryos (*Figure 3E,F*). Notably, penetrant division failure was observed in the embryos in the uterus in the upshifted *cyk-4*(*t1689ts*) and *zen-4*(*or153ts*) mutant worms, confirming that the interaction between the CYK-4 and ZEN-4 dimers is essential for cytokinesis (*Figure 3E*). Thus, analysis of germline structure and embryo production following RNAi-mediated depletion of CYK-4 and ZEN-4 beginning at the L4 stage as well as following upshift of fast-acting temperature-sensitive mutants at the L4 stage indicates that CYK-4 plays a critical role in germline structure and embryo production in the adult, whereas ZEN-4 does not.

To assess the role of the interaction between the CYK-4 and ZEN-4 in development, we upshifted the centralspindlin assembly mutants at the L1 larval stage and analyzed them 24 hr later at the L4 stage. In contrast to the mutant that disrupts the CYK-4 C-terminus (*cyk-4(or749ts)*), the upshifted centralspindlin assembly mutants (*cyk-4*(*t1689ts*) and *zen-4*(*or153ts*)) exhibited a global growth defect (*Figure 3—figure supplement 1A*). Even following a subsequent downshift for 24 hr, the



**Figure 2.** CYK-4 is required for oocyte production by the adult germline. (**A**) The CYK-4 C-terminus is not required for compartment proliferation during germline development. (*Upper left*) Larvae were upshifted to the non-permissive temperature (25°C) at the L1 stage and the germlines were examined at the L4 stage as indicated in the schematic. (*Upper right*) Representative single plane confocal images of germlines in L4 stage worms (control or *cyk-4(or749ts)*) expressing a GFP-tagged plasma membrane probe (shown in red) and mCherry::histone H2B (shown in green) after the upshift protocol. Dashed lines mark the germline boundaries. (*Lower left*) Schematic outline of the upshift protocol used to assess embryo production. (*Lower right*) Graph plots the number of embryos laid by the worms between 24 and 48 hr after downshift. (**B**) The CYK-4 C-terminus is required for oocyte production in the adult. (*Upper left*) Larvae were upshifted to the non-permissive temperature (25°C) at the L4 stage and the germlines were examined 28 hr later as indicated in the schematic. (*Upper right*) Representative single plane confocal images of germlines in adult stage worms (control or *cyk-4 (or749ts)*) expressing a GFP-tagged plasma membrane probe (shown in red) and mCherry::histone H2B (shown in green) after the upshift. (*Lower left*) Schematic outline of the upshift protocol used to assess embryo production. (*Lower right*) Graph plots the number of embryos produced between 24 and 48 hr after upshift. n in images in A and B = number of imaged germlines. N in graphs = number of assayed worms. Scale bars are 10 μm.

DOI: https://doi.org/10.7554/eLife.36919.008

The following source data is available for figure 2:

**Source data 1.** The CYK-4 C-terminal region is not required for germline compartment proliferation.

DOI: https://doi.org/10.7554/eLife.36919.009

*Figure 2 continued on next page*

*Figure 2 continued*

**Source data 2.** The CYK-4 C-terminal region is required for oocyte production.
DOI: https://doi.org/10.7554/eLife.36919.010

resulting adults remained significantly smaller than comparably treated control or *cyk-4(or749ts)* worms (*Figure 3—figure supplement 1A*). The germlines in the centralspindlin assembly mutant worms that were upshifted between the L1 and L4 stages were also smaller and exhibited apparent 'pathfinding' defects (*Figure 3—figure supplement 1B*). The aberrant germlines in the L1-L4 upshifted centralspindlin assembly mutants were also not able to produce embryos following temperature downshift (*Figure 3—figure supplement 1C*), in agreement with prior work that reported germline defects and sterility in worms subjected to *zen-4* RNAi beginning at the L1 stage (*Zhou et al., 2013*). The global effect on worm growth of the centralspindlin assembly mutants makes it difficult to interpret the germline phenotypes, because they likely arise as a secondary consequence. We note that, despite the global inhibition of growth, the germlines exhibited compartment proliferation, and multinucleated compartments were not observed. The essential role of centralspindlin in worm growth may reflect a microtubule-related function because the *cyk-4 (or749ts)* mutant that fails to promote cytokinesis, but retains the ability to interact with ZEN-4 and can promote central spindle assembly (*Canman et al., 2008*) is able to support normal worm growth.

We conclude that the interaction between CYK-4 and ZEN-4, which is required for central spindle assembly during cytokinesis (*Jantsch-Plunger et al., 2000*; *Mishima et al., 2002*; *Severson et al., 2000*), is essential for worm growth/development between the L1 and L4 stages, whereas the CYK-4 C-terminus is not. Conversely, the CYK-4 C-terminus, which is required for contractile ring assembly (*Canman et al., 2008*; *Davies et al., 2014*; *Loria et al., 2012*; *Zhang and Glotzer, 2015*), is essential for oocyte production in adult worms, whereas ZEN-4 is not.

One reason for why ZEN-4, a microtubule motor, could be dispensable for embryo production in the adult germline is suggested by imaging of a strain expressing fluorescent β-tubulin (*Figure 3G*). As previously reported, abundant microtubules are present throughout the germline, extending into compartments and oocytes (*Wang et al., 2015*; *Wolke et al., 2007*; *Zhou et al., 2013*). However, in contrast to cytokinesis, where contractile ring assembly is directed by an organized set of anti-parallel microtubule bundles called the central spindle, the rachis bridges in the oocyte cellularization region lack an organized microtubule-based structure (see also [*Wolke et al., 2007*]). In addition, depletion of the PRC1 homolog SPD-1, a microtubule bundling protein required for central spindle assembly (*Verbrugghe and White, 2004*), did not lead to any apparent changes in microtubule organization or germline architecture (*Figure 3G*). We conclude that in contrast to cytokinesis, where both CYK-4 and ZEN-4 are required, CYK-4 can act independently of ZEN-4 to support oocyte production. We suggest that the differential importance of ZEN-4 in cytokinesis compared to oocyte cellularization may reflect a difference in the role of microtubule-based signaling in the two contexts.

## CYK-4 is required for oocyte cellularization

Our results indicated that depleting CYK-4, or disrupting the C-terminal region containing its GAP and C1 domains using a mutant, leads to a tubulated proximal germline that lacks compartment boundaries (*Figure 2B*, *Figure 3A*). To understand the origin of this phenotype, we developed the means to longitudinally monitor individual living worms over a 5-hr period using a specially made microfluidic device ('worm trap'; *Figure 4A*, *Figure 4—figure supplement 1*), in which worms are periodically immobilized for imaging and then released to move and feed between timepoints (spaced 50–75 min apart). Bacterial suspension is perfused through the microchannel network throughout the imaging session to enable feeding. This setup is conceptually similar to a recently presented microfluidic platform for high-resolution longitudinal imaging of *C. elegans* (*Keil et al., 2017*); however, the device we designed is simpler and easier to make, because it does not have an embedded microchannel layer. The loading of worms into the device, which is accomplished by manually transferring them into a drop of liquid on top of the device using a standard platinum worm pick, is also straightforward.

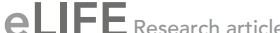

**Figure 3.** ZEN-4 is not essential for embryo production by the adult syncytial germline. (**A**) (*Top*) Schematic outline of the imaging experiment following dsRNA injection. (*Bottom*) Single plane confocal images of adult germlines expressing a GFP-tagged plasma membrane probe (shown in red) and mCherry::histone H2B (shown in green) following depletion of CYK-4 or ZEN-4 by RNAi. n = number of worms imaged. (**B**) Graphs plotting the number of embryos laid by the worms during the indicated time intervals (*left*) or embryonic viability (*right*) (mean ± SD) for the indicated conditions.

*Figure 3 continued*

N = number of worms, n = number of embryos. (**C**) Maximum intensity projections of the adult germline in control and *zen-4(RNAi)* worms expressing *in situ*-tagged GFP::ZEN-4 and an mCherry-tagged plasma membrane probe. n = number of worms imaged. (**D**) (*left three columns*) Maximum intensity projection images of adult germlines in control (*top*) and *zen-4(RNAi)* (*bottom*) worms expressing CYK-4::mNeonGreen (*green*) and a mCherry-tagged plasma membrane probe (*red*). (*right column*) Single plane images of a portion of the pachytene region. n = number of imaged worms. (**E**) (*Top*) Schematic outline of the experiment imaging adult worms following temperature upshift at the L4 stage. (*Bottom*) Single plane confocal images showing the GFP-tagged plasma membrane probe in adult germlines and embryos in the uterus of worms subjected to the upshift protocol. n = number of worms imaged. (**F**) Graph plotting the number of embryos produced (between 24 and 48 hr after temperature upshift at the L4 stage. Embryos in the uterus were counted in addition to embryos laid because centralspindlin mutants, especially *cyk-4(t1689ts)*, are somewhat egg laying defective under these conditions. (**G**) Images of adult germlines in control (*left*) and *spd-1(RNAi)* (*right*) worms expressing GFP::β-tubulin, mCherry::histone and an mCherry-tagged plasma membrane probe. Insets (bottom panels) show the GFP::β-tubulin signal in the boxed regions (oocyte cellularization region) with yellow dashed lines outlining the cell boundaries). n = number of imaged worms. As illustrated in (**A**), after dsRNA injection, worms were incubated for 48 hr at 20°C before imaging in (**C**), (**D**) and (**G**). Scale bars are 10 μm.

DOI: https://doi.org/10.7554/eLife.36919.011

The following source data and figure supplements are available for figure 3:

**Source data 1.** ZEN-4 is required for embryo viability but not production.

DOI: https://doi.org/10.7554/eLife.36919.013

**Source data 2.** The CYK-4—ZEN-4 interaction is not required for embryo production.

DOI: https://doi.org/10.7554/eLife.36919.014

**Figure supplement 1.** The interaction between the CYK-4 and ZEN-4 dimers is globally required for worm growth and development between the L1 and L4 stages.

DOI: https://doi.org/10.7554/eLife.36919.012

**Figure supplement 1—source data 1.** The CYK-4—ZEN-4 interaction is required for larval growth and germline pathfinding.

DOI: https://doi.org/10.7554/eLife.36919.015

**Figure supplement 1—source data 2.** Aberrant germlines in L1-L4 upshifted centralspindlin assembly mutants fail to produce embryos following downshift.

DOI: https://doi.org/10.7554/eLife.36919.016

To determine how the tubulated proximal germline phenotype arises following CYK-4 inhibition, we combined the worm trap imaging approach with temperature upshift of the *cyk-4(or749ts)* mutant that disrupts the C1-GAP domain module in the CYK-4 C-terminus. As a control, we conducted the same analysis with the *cyk-4(t1689ts)* mutant, which disrupts the CYK-4—ZEN-4 interaction but does not significantly affect germline structure. Upshifting adult *cyk-4(t1689ts)* mutant worms did not prevent oocyte cellularization during the 5 hr imaging period. In contrast, upshifting the *cyk-4(or749ts)* mutant, resulted in gradual development of the tubulated germline phenotype. The phenotype initiated at the proximal tip of the germline where the nascent oocyte was in the process of cellularizing to separate from the germline syncytium. In normal germlines, the diameter of the rachis decreases, tapering to a pointed tip, in parallel with the reduction in the diameter of the rachis bridges that open into the nascent oocytes. The combination of these two events brings the distal partition of the oocyte up to the top of the germline tube and cellularizes the oocyte to bud it off the rachis tip (*Figure 1C*, *Figure 4B,C*). In *cyk-4(or749ts)* germlines after the temperature upshift, the rachis tip above the cellularizing oocyte appeared to widen and lose its structure (*Figure 4B*, yellow arrowheads in 50 and 110 min timepoints), suggesting that it no longer tapered normally; at the same time, the rachis bridges in the cellularizing oocyte and adjacent compartment opened up (*Figures 4B*, 50 and 110 min timepoints). Together, these events caused retraction of the partition separating the last uncellularized oocyte from its neighbor (*Figure 4B*, white arrowheads in 50 to 185 min timepoints, *Figure 4C*). This was followed by opening of the bridge to the next compartment and loss of another partition (purple arrowheads in 185 and 245 min timepoints).

We also analyzed the width of rachis bridges in the pachytene region after 5–6 hr at non-permissive temperature (*Figure 4—figure supplement 2*) and found that the width of the rachis bridges increased in both control and *cyk-4(or749ts)* worms after a 5.5 to 6 hr upshift to 25°C. The increase was greater in *cyk-4(or749ts)* worms than in controls so that the average pachytene rachis bridge diameter was ~4.5 ± 0.8 μm in *cyk-4(or749ts)* worms compared to 3.7 ± 0.5 μm in controls at 25°C. It is unclear if this increase in rachis bridge diameter arises due to a mild reduction in rachis contractility or as an indirect consequence of the disruption of oocyte cellularization. Although it is possible that the failure of oocyte cellularization in the upshifted mutants results from the fact that bridge

**Figure 4.** CYK-4 is required for oocyte cellularization. Individual CYK-4 mutant worms were longitudinally monitored using a custom vacuum-actuated microfluidic device (the 'worm trap'). (**A**) Schematics summarize the experimental procedure for mounting worms in the trap (*top*), temperature shift (*middle*), and periodic immobilization for imaging (*bottom*). Shown are a series of images demonstrating gradual immobilization and release of a single worm at low magnification (10x), and a single plane high-resolution (60x) image of immobilized worm. (**B**) (*top*) Individual young adult worms carrying

*Figure 4 continued on next page*



*Figure 4 continued*

the *cyk-4(t1689ts)* mutation that blocks the interaction between CYK-4 and ZEN-4 dimers (n = 7; Centralspindlin Assembly) or the *cyk-4(or749ts)* mutation that compromises the CYK-4 C-terminus (n = 9; CYK-4 C-term) expressing a GFP-tagged plasma membrane probe (shown in red) and an mCherry fusion with histone H2B (shown in green) were tracked for 5 hr at restrictive temperature in the worm trap. Worms were immobilized periodically for imaging, as indicated in the schematic in (A). Single plane images from the time course are shown. (C) Schematics illustrate how the phenotype in *cyk-4(or749ts)* worms arises. Following upshift, the partition between the last two uncellularized oocytes resects due to the combined effects of the rachis widening and the rachis bridges opening up (for example, see white arrowheads in the sequence in (B)). Resection of the next and subsequent partitions in a similar fashion (purple arrowheads in the sequence in (B)) leads to a hollow multinucleated 'tubulated' proximal germline. Scale bars are 10 μm.

DOI: https://doi.org/10.7554/eLife.36919.017

The following source data and figure supplements are available for figure 4:

**Figure supplement 1.** Design of the worm trap microfluidic chip.

DOI: https://doi.org/10.7554/eLife.36919.018

**Figure supplement 2.** Measurement of the rachis bridges in the CYK-4 C-terminal mutant *cyk-4(or749ts)*.

DOI: https://doi.org/10.7554/eLife.36919.019

**Figure supplement 2—source data 1.** Rachis bridges in the pachytene region are wider in upshifted *cyk-4(or749ts)* mutant worms than in controls.

DOI: https://doi.org/10.7554/eLife.36919.020

width in the pachytene region is too large, we think this is unlikely, as rachis bridges in wild-type germlines are often observed to expand beyond a diameter of 4.5 μm as they are loaded with components in the turn region (*Figure 1C*).

We conclude, based on longitudinal imaging with a worm trap after temperature upshift, that the germline defect in the *cyk-4(or749ts)* mutant arises in a proximal to distal fashion due to failure of the two coordinated events that normally cellularize oocytes: compartment bridge closure and tapering of the rachis. Thus, CYK-4, and specifically its C-terminal C1-GAP module, is critical for the bridge closure that cellularizes oocytes to separate them from the germline syncytium.

## CYK-4 requires both its C1 domain and the Rho GTPase interaction interface of its GAP domain to localize to the rachis surface and bridges and to promote oocyte cellularization

The data above indicate that the C-terminal region of CYK-4 containing the C1 and GAP domains is essential for CYK-4 function in the germline. Prior work has suggested that the CYK-4 C1 domain is required to target CYK-4 to the plasma membrane and that blocking C1 domain-mediated membrane targeting leads to a loss-of-function phenotype (*Basant et al., 2015*; *Zhang and Glotzer, 2015*). The *cyk-4(or749ts)* mutant is thought to disrupt the function of the C1 as well as the GAP domain (*Zhang and Glotzer, 2015*), precluding assessment of the role of each domain. To analyze the respective roles of the C1 and GAP domains, we therefore generated a set of RNAi-resistant transgenes under the endogenous *cyk-4* promoter encoding untagged wild-type CYK-4 (WT), CYK-4 with the C1 domain deleted (ΔC1), and CYK-4 mutants with changes predicted based on prior structural work to disrupt GAP activity (R459A; mutation of the 'arginine finger') and the ability of the GAP domain to interact with Rho family GTPases (AAE mutant; R459A, K495A and R499E; *Figure 5A*, *Figure 5—figure supplement 1A*). We note that R459 is part of the GTPase interacting interface, and we cannot exclude the possibility that its mutation could affect GTPase binding as well as GAP activity. All transgenes were expressed at levels comparable to endogenous CYK-4 (*Figure 5B*, *Figure 5—figure supplement 1B*). Examination of germline structure and counting the number of embryos laid 24–48 hr after injection of dsRNA to deplete endogenous CYK-4 confirmed that the transgene encoding WT CYK-4 rescued both germline structure and embryo production. In contrast, the ΔC1, R459A and AAE mutants all failed to rescue (*Figure 5C*, *Figure 5—figure supplement 1C*). Thus, both the C1 and the GTPase binding interface of the GAP domain are important for the germline function of CYK-4.

To examine the effects of the engineered mutations on the ability of CYK-4 to localize to the rachis surface, we generated a parallel series of RNAi-resistant transgenes with a C-terminal mCherry tag (*Figure 5—figure supplement 1D*). WT CYK-4::mCherry localized to the rachis surface and was enriched in the rachis bridges; it was also present in nuclei, similar to mNeonGreen-tagged CYK-4 (*Figure 5D*, *Figure 1B*). In contrast, the ΔC1, R459A and AAE mutants failed to localize to the rachis

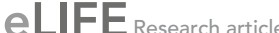

**Figure 5.** CYK-4 requires both its C1 domain and the Rho GTPase interaction interface of its GAP to target to the rachis surface and bridges and promote oocyte cellularization. (**A**) (*Left*) Schematic illustrates the set of single copy untagged RNAi-resistant *cyk-4* transgenes inserted into a specific location on Chr. II generated to analyze the role of the C1 and GAP domains. (*Right*) Structure of the complex of human Cyk4 GAP domain (grey) and RhoA (blue) (reproduced based on PBD ID 5C2K). Key residues in the binding pocket (numbered as per *C. elegans* CYK-4) are highlighted in orange

*Figure 5 continued on next page*

*Figure 5 continued*

(R459, catalytic arginine finger) and magenta (K495 and R499, important for GTPase binding). (**B**) Immunoblots of extracts prepared from worms lacking a transgene (None; N2 strain) or with the transgenes outlined in (**A**) in the presence (+) or absence (-) of RNAi to deplete endogenous CYK-4. Blots were probed with antibodies to CYK-4 (*top*) and α-tubulin as a loading control (*bottom*). With the exception of the ΔC1 mutant, which runs at a lower molecular weight, the transgene-encoded proteins ran at the same molecular weight as endogenous CYK-4. In the absence of a *cyk-4* transgene, the CYK-4 band disappears, confirming the effectiveness of our RNAi. The protein running at the level of endogenous CYK-4 after RNAi in the WT, R459A, AAE samples is the transgenic protein. (**C**) (*Left*) Single central plane images of the germline in adult worms with the indicated transgenes that were also expressing mCherry::histone and an mCherry plasma membrane maker after depletion of endogenous CYK-4 by RNAi. n = number of worms. (*Right*) Graph plotting the number of embryos laid 24–48 hr post-injection (mean ± SD) for strains expressing the indicated *cyk-4* transgenes (without histone or plasma membrane markers). N = number of worms. (**D**) Single central plane images of the germline in adult worms expressing the indicated CYK-4::mCherry fusions. Asterisks highlight the localization of each of the fusions to nuclei and arrows point to the rachis surface in each germline. n = number of imaged worms. (**E**) Immunoblot of extracts prepared from worms expressing the mCherry-tagged RNAi-resistant *cyk-4* transgenes probed with antibodies to CYK-4 (*top*) and α-tubulin (*bottom*) as a loading control. Scale bars are 10 μm.

DOI: https://doi.org/10.7554/eLife.36919.021

The following source data and figure supplements are available for figure 5:

**Source data 1.** Both the C1 domain and the GTPase binding interface of the GAP domain are required for the germline function of CYK-4.

DOI: https://doi.org/10.7554/eLife.36919.023

**Figure supplement 1.** The WT, GTPase binding interface mutant, and ΔC1 mutant CYK-4 proteins encoded by the single copy untagged transgenes are expressed at levels comparable to endogenous CYK-4.

DOI: https://doi.org/10.7554/eLife.36919.022

**Figure supplement 1—source data 1.** Both the C1 and the GTPase binding interface of the GAP domain are required for embryonic viability.

DOI: https://doi.org/10.7554/eLife.36919.024

**Figure supplement 1—source data 2.** The RNAi-resistant transgene encoding CYK-4::mCherry rescues depletion of endogenous CYK-4.

DOI: https://doi.org/10.7554/eLife.36919.025

surface and rachis bridges, although they were present at levels comparable to the WT protein in nuclei (*Figure 5D*) and localized to central spindles in embryos (*not shown*). Western blotting further confirmed comparable expression of the mCherry-tagged WT and mutant proteins (*Figure 5E*). These results suggest that both the C1 domain and the Rho GTPase-binding interface are essential for the targeting of CYK-4 to the rachis surface and rachis bridges. We note that these experiments were performed in the presence of endogenous CYK-4, with which the mutant proteins are expected to dimerize via their N-terminal coiled coil, and in the presence of endogenous ZEN-4. Thus, these results raise the possibility that centralspindlin complexes containing even one mutant C1 or one mutant GAP domain are unable to localize to the rachis surface. We conclude that the C1 domain that binds lipids and the GAP domain interface that interacts with Rho family GTPases are both required for the recruitment of CYK-4 to the rachis surface and for oocyte cellularization.

## The Rho GTPase-binding interface of the CYK-4 GAP domain may recruit CYK-4 to the rachis surface/bridges by binding RhoA[RHO-1]

Our results indicated that both the C1 domain of CYK-4 and the Rho GTPase binding interface of its GAP domain are required for CYK-4 to localize to the rachis surface and rachis bridges in the germline. In contrast to a prior report (*Zhou et al., 2013*), depletion of the germline anillin homolog ANI-2, which alters germline structure, did not prevent CYK-4 localization (*Figure 6—figure supplement 1*). To understand how CYK-4 is recruited to the rachis surface/bridges, we began by analyzing the dynamics of this CYK-4-containing structure. To this end, we bleached a section of the rachis in the pachytene region in anesthetized adult worms expressing CYK-4::mNeonGreen and imaged the central-most plane. No detectable recovery of the bleached region was observed over a 5-min interval following the bleach (*Figure 6A*), suggesting that the CYK-4 containing structure that lines the rachis surface/bridges is not rapidly turning over.

Since the localization of CYK-4 to the rachis surface/bridges depends on the Rho GTPase-binding interface of its GAP domain, we tested the role of Rho family GTPases in CYK-4 localization. As a first step, we counted the number of embryos laid 24–48 hr after injection of dsRNA targeting RhoA (RHO-1 in *C. elegans*), Rac (CED-10 in *C. elegans*; this RNA likely also targets the second *C. elegans* Rac homolog RAC-2) and Cdc42 (CDC-42). Inhibition of Rac[CED-10] or CDC-42 did not affect embryo production (*Figure 6B*) or the localization of CYK-4 to the rachis surface/bridges (*Figure 6C*);

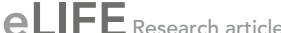

**Figure 6.** The Rho GTPase binding interface of the CYK-4 GAP domain may recruit CYK-4 to the rachis surface/bridges by binding RhoA$^{RHO-1}$. (**A**) Central plane confocal images acquired in adult worms expressing CYK-4::mNeonGreen (*green*) and an mCherry-tagged plasma membrane probe (*red*) before and after photobleaching of the CYK-4::mNeonGreen signal in a section of the rachis in the pachytene region. (**B**) (*Left*) Single central plane images of germlines in adult worms expressing mCherry::H2B (shown in green) and a GFP-tagged plasma membrane probe (shown in red) following CYK-4 or RhoA$^{RHO-1}$ depletion. n = number of worms. (*Right*) Plot of number of embryos laid by the worms 24–48 hr after dsRNA injection (mean ± SD) for the indicated targets in the N2 background. N = number of worms. (**C**) Single central plane images of germlines in adult worms expressing CYK-4::mCherry (shown in green) and a GFP-tagged plasma membrane probe (shown in red) following depletion of Rac$^{CED-10}$ or Cdc42$^{CDC-42}$ (48 hr post-injection). (**D**) Single central plane images of germlines in adult worms expressing CYK-4::mNeonGreen (*green*) and an mCherry-tagged plasma membrane probe (*red*) following partial depletion of RhoA$^{RHO-1}$ or ECT-2 (20 hr post-injection). Insets (*bottom*) show the boxed regions magnified 1.5x. n = number of worms.

DOI: https://doi.org/10.7554/eLife.36919.026

The following source data and figure supplements are available for figure 6:

**Source data 1.** Depletion of RhoA or CYK-4 leads to a comparable reduction in embryo production.
DOI: https://doi.org/10.7554/eLife.36919.030

**Figure supplement 1.** The anillin homolog, ANI-2, is not required to recruit CYK-4 recruitment to the rachis surface/bridges.
DOI: https://doi.org/10.7554/eLife.36919.027

**Figure supplement 2.** Depletion of Rac$^{CED-10}$ or CDC-42 cannot rescue the effects of mutants in the Rho GTPase binding interface on the germline.
DOI: https://doi.org/10.7554/eLife.36919.028

*Figure 6 continued on next page*

*Figure 6 continued*

**Figure supplement 2—source data 2.** Depletion of Racor CDC-42 does not rescue the germline defects of the CYK-4 Rho GTPase binding interface mutants.

DOI: https://doi.org/10.7554/eLife.36919.029

depletion of Rac[CED-10] or CDC-42 also did not rescue the germline defects of the CYK-4 Rho GTPase-binding interface mutants (*Figure 6—figure supplement 2*). In contrast, RhoA[RHO-1] depletion reduced embryo production and led to a tubulated germline phenotype similar to that resulting from CYK-4 depletion (*Figure 6B*). Because the rachis surface and bridges are lost following penetrant depletion of RhoA[RHO-1] or its GEF, ECT-2, the amount of CYK-4 on the rachis surface cannot be analyzed. We therefore partially depleted ECT-2 and RhoA[RHO-1] and analyzed CYK-4 localization. Levels of CYK-4::mNeonGreen on the rachis surface/bridges were qualitatively reduced by both depletions (*Figure 6D*). However, this experiment does not distinguish between the reduction in CYK-4 levels occurring because an interaction between the CYK-4 GAP domain and RhoA[RHO-1] is required for the initial localization of CYK-4 or for its stable maintenance. We also cannot rule out the possibility that the Rho GTPase binding interface of the CYK-4 GAP domain is required for CYK-4 deposition via an interaction with an as yet uncharacterized protein, and the reduction in CYK-4 levels when RhoA[RHO-1] is depleted is an indirect consequence of RhoA[RHO-1] depletion compromising the structure of the rachis lining/bridges. However, we think this is unlikely, particularly since the set of proteins required for germline function has been comprehensively characterized (*Green et al., 2011*) and did not yield an attractive candidate whose inhibition gives a CYK-4 like phenotype.

In summary, both the CYK-4 C1 domain and the interface of the CYK-4 GAP domain predicted to interact with Rho family GTPases are required for incorporation of CYK-4 into the structure that lines the rachis surface and rachis bridges in the germline. Although we cannot rule out the possibility that the Rho GTPase-binding interface of the CYK-4 GAP domain is required for CYK-4 to accumulate at this site via an interaction with an as yet uncharacterized protein, we think it is most likely that it mediates recruitment by binding to RhoA[RHO-1].

## Active RhoA[RHO-1] localizes to the rachis surface/bridges in an ECT-2 and CYK-4-dependent manner

During conventional cytokinesis, both in *C. elegans* and in other systems, CYK-4 contributes to RhoA activation (*White and Glotzer, 2012*; *Zhang and Glotzer, 2015*). To determine if CYK-4 also contributes to RhoA activation in the *C. elegans* germline, we expressed a GFP fusion with RGA-3, the major GAP that targets RhoA in the germline and in embryos (*Green et al., 2011*; *Schmutz et al., 2007*; *Schonegg et al., 2007*; *Zanin et al., 2013*). Our prior work suggested that RGA-3 localization follows the localization of active RhoA (*Zanin et al., 2013*). Consistent with the idea that RhoA[RHO-1] is activated on the rachis surface, GFP::RGA-3 concentrated on the rachis surface/bridges, and this localization was lost following partial inhibition of RhoA[RHO-1] or its activating GEF, ECT-2 (*Figure 7A*). Unlike CYK-4::mNeonGreen, GFP::RGA-3 on the rachis surface/bridges recovered following bleaching (*Figure 7B*), suggesting that active RGA-3-accessible RhoA[RHO-1] turns over more rapidly than the CYK-4 containing structure that lines the rachis. We note that if there is a stable population of RhoA[RHO-1] that is incorporated along with CYK-4 into the rachis/bridge lining, it would not be detected with either our GFP::RGA-3 probe or with the C-terminal fragment of anillin that has previously been used to visualize active RhoA (*Tse et al., 2012*), since steric constraints would prevent both probes from simultaneously associating with GAP domain-bound RhoA. Consistent with a role for CYK-4 in activating RhoA in the germline, partial depletion of CYK-4 under conditions where the structure of the germline remained largely intact, led to loss of GFP::RGA-3 from the rachis surface (*Figure 7A*). During embryonic cytokinesis, mutations in the Rho GTPase binding interface of CYK-4 have been shown to compromise RhoA[RHO-1] activation by CYK-4. In this context, inhibition of the paralogous RhoA GAPs RGA-3 and RGA-4 can partially suppress the resulting cytokinesis defects, suggesting that the mutants retain some ability to reach the plasma membrane and activate RhoA during cytokinesis. In the germline, RGA-3/4 depletion could not rescue the effect of the R459A mutation on embryo production (*Figure 7C*), which is consistent with our observation

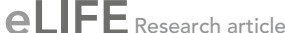

**Figure 7.** Active RhoA$^{RHO-1}$ localization to the rachis surface/bridges depends on ECT-2 and CYK-4. (**A**) Maximum intensity projections (*top*) and single central plane images (*middle*) of the germline in adult worms expressing GFP::RGA-3 (*green*) and an mCherry-tagged plasma membrane probe (*red*) following partial depletion of RhoA$^{RHO-1}$, ECT-2, or CYK-4 (20 hr post-injection). Insets (*bottom*) show the boxed regions magnified 1.5x. n = number of worms. (**B**) Central plane confocal images acquired in adult worms expressing GFP::RGA-3 (*green*) and an mCherry-tagged plasma membrane probe (*red*) before and after photobleaching of the GFP::RGA-3 signal in a section of the rachis in the pachytene region. (**C**) Graph plotting the number of embryos laid (mean ± SD) by worms expressing wildtype or R459A CYK-4::mCherry from RNAi-resistant transgenes following depletion of endogenous CYK-4 and/or RGA-3/4 as indicated. N = number of worms. Scale bars are 10 µm.

DOI: https://doi.org/10.7554/eLife.36919.031

*Figure 7 continued on next page*

*Figure 7 continued*

The following source data is available for figure 7:

**Source data 1.** RGA-3/4 depletion cannot rescue the effect of the R459A mutation on embryo production.

DOI: https://doi.org/10.7554/eLife.36919.032

that the R459A mutant protein cannot localize to the rachis surface and is therefore unlikely to retain function.

## Discussion

During gametogenesis in metazoans, germ cells often undergo incomplete cytokinesis to form syncytia connected by intercellular bridges. In female germlines, syncytial organization facilitates oocyte provisioning (*King and Mills, 1962*; *Lei and Spradling, 2016*; *Pepling, 2016*; *Robinson and Cooley, 1997*); however, it also necessitates a mechanism for closing intercellular bridges to cellularize oocytes prior to fertilization, a process about which relatively little is known. The most conserved component of intercellular bridges, observed in all bridges examined, is the centralspindlin complex (*Carmena et al., 1998*; *Greenbaum et al., 2009*, *2007*, *2006*; *Haglund et al., 2010*; *Minestrini et al., 2002*; *Zhou et al., 2013*). Surprisingly, given the equivalent functional importance of the two centralspindlin subunits during cytokinesis, we show that the CYK-4 GAP subunit of centralspindlin, but not the ZEN-4 motor, is required for the bridge closure that cellularizes oocytes in *C. elegans*. Engineered mutants revealed that the Rho GTPase-binding interface of the CYK-4 GAP domain collaborates with the lipid-binding C1 domain to target CYK-4 to intercellular bridges (see model in *Figure 8A*). This leads us to propose that the conserved C1-GAP region of CYK-4 constitutes a targeting module that functions in a fashion similar to the mechanism recently shown to recruit anillin to the cortex, in which its C2, Rho-binding, and PH domains all form low-affinity interactions that collectively target anillin to the membrane (*Sun et al., 2015*).

### A potential feedback loop for RhoA activation involving CYK-4 GAP domain-mediated localization

Our analysis revealed that GFP::RGA-3, which reads out a population of active RhoA, is lost on the rachis surface and bridges following CYK-4 depletion, suggesting that CYK-4 promotes RhoA activation in the germline. This finding is consistent with prior work on CYK-4 suggesting that it promotes RhoA activation during embryonic cytokinesis (*Loria et al., 2012*; *Zhang and Glotzer, 2015*). How CYK-4 contributes to RhoA activation in either context is not known, although this function may require the region between its coiled-coil and C1 domains that, in human cells, directly binds to Ect2 (*Burkard et al., 2009*; *Wolfe et al., 2009*). Much attention has been focused on the role of the CYK-4 GAP domain, since it is counter-intuitive that a GAP targeting Rho family GTPases would be required to activate RhoA. Our data in the *C. elegans* germline suggests that the Rho GTPase-binding interface of the GAP domain collaborates with the C1 domain, most likely by binding membrane-localized RhoA, to target CYK-4 to the rachis surface and rachis bridges. Thus, mutations in the Rho-binding interface of the GAP domain disrupt CYK-4 localization to the rachis surface and mimic loss of RhoA function in the germline (*Figure 5C,D*; *Figure 6B*; [*Green et al., 2011*]). Recent work on embryonic cytokinesis also found that mutations in the Rho GTPase-binding interface reduced RhoA activation (*Zhang and Glotzer, 2015*). The authors in that study proposed an alternative model in which the CYK-4 GAP domain interacts with the GEF domain of ECT-2, in a RhoA-dependent fashion, to activate it. However, it is important to note that this remains a speculative suggestion, as in vitro experiments have so far failed to support the idea that the CYK-4 GAP domain can activate the ECT-2 GEF (*Zhang and Glotzer, 2015*). Additional work will be needed to determine if the C1-GAP region of CYK-4 contributes to targeting CYK-4 to the cortex during cytokinesis as it does in the germline, or if it promotes RhoA activation via a distinct mechanism in this context.

Our results suggest that CYK-4 that is stably associated with the rachis surface contributes to the generation of a dynamic population of active RhoA (RGA-3-accessible; *Figure 7A*). The population of active RhoA generated by CYK-4 may, in turn, enable the recruitment of additional CYK-4 via its



**Figure 8.** A C-terminal C1 domain-GAP module targets CYK-4 to the rachis surface/bridges to enable oocyte celluarization. (**A**) Schematics compare the closure of compartment bridges during oocyte cellularization in the germline (*left*) to cytokinesis (shown here in a two-cell stage embryo, *right*). (*Left*) During bridge closure in the germline, the CYK-4 C1 domain and the Rho GTPase binding interface collaborate to recruit CYK-4 to the rachis surface/bridges. In the germline, bridges are not bisected by a central spindle-like microtubule bundles and the kinesin-6, ZEN-4, although present, is not essential for CYK-4 targeting or cellularization. (*Right*) During cytokinesis, cortical contractility is patterned by the anaphase spindle. Both ZEN-4 and CYK-4 are required to assemble the central spindle and contractile ring. Targeting to the central spindle and other ZEN-4/microtubule-dependent mechanisms may help deliver CYK-4 to the cell surface. We speculate that the C-terminal module-based mechanism that operates in the germline may also contribute to targeting CYK-4 to the plasma membrane during cytokinesis. (**B**) Our data suggest that CYK-4 associated with the rachis surface promotes RhoA activation. If active RhoA$^{RHO-1}$ generated by membrane-associated CYK-4 enables the recruitment of additional CYK-4 via interaction with its C1-GAP module, this would result in a positive feedback that promotes RhoA activation and CYK-4 recruitment.

DOI: https://doi.org/10.7554/eLife.36919.033

C1-GAP module, leading to a positive feedback loop that promotes RhoA activation *Figure 8B*. Such feedback would need to be restricted, potentially by the antagonistic activity of localized RGA-3/4, to achieve a specific level of RhoA activation. This feedback model could explain why the

disruption of localization by selective mutations in the C1 or GTPase-binding interface of CYK-4, mimics RhoA loss-of-function in the germline.

## Mechanisms triggering closure of intercellular bridges in the proximal germline

Our results suggest that a feedback loop involving CYK-4 mediated by its C-terminal C1-GAP module may ensure sufficient RhoA activation to enable closure of intercellular bridges and oocyte cellularization. However, given that RGA-3-accessible active RhoA is present on the rachis surface/bridges throughout the germline, an important question that remains is why bridge closure is specifically triggered in the proximal region of the germline. In dividing cells, a key regulatory event that promotes cytokinesis is reduction of Cdk1 kinase activity due to the activation of the anaphase-promoting complex/cyclosome (APC/C), the E3 ubiquitin ligase that targets cyclin B for degradation (*Green et al., 2012*). The cytokinesis-like event that cellularizes oocytes occurs near the end of meiotic prophase, prior to nuclear envelope breakdown and the first round of meiotic chromosome segregation. How or whether cell cycle regulators bring about oocyte expansion and cellularization is an open question. Consistent with the idea that oocyte cellularization could also be triggered by APC/C activation, we previously identified APC/C subunits in a phenotypic class characterized by the accumulation of oocytes with large open rachis bridges and compromised rachis tapering (*Green et al., 2011*). More work will be needed to determine whether transient APC/C activation is the trigger for cortical constriction during oocyte cellularization, as it is during cytokinesis, and, if so, to identify the target whose degradation promotes bridge closure.

## Potential reasons why oocyte cellularization does not require ZEN-4

Our data suggest that the ZEN-4 subunit of centralspindlin is not required to promote the closure of rachis bridges that cellularizes oocytes to separate them from the germline syncytium. This conclusion is supported by our analysis of germline structure and embryo production following RNAi-mediated depletion of CYK-4 and ZEN-4, as well as following upshift of temperature-sensitive mutants, both beginning at the L4 stage. We note that our conclusion is in contrast to a prior report based on analysis of fixed extruded germlines that suggested that *cyk-4*(*or749ts*) and *zen-4*(*or153ts*) germlines exhibit a similar phenotype (*Zhou et al., 2013*). A direct comparison with the prior work is difficult because the nature of the defects in the mutant germlines were not quantified; the images shown highlighted small patches of multinucleated compartments in the pachytene region of germlines. Although we have observed bi-nucleate compartments in the pachytene region in upshifted *zen-4 (or153ts)* worms (particularly older worms), these events were rare. In contrast to this subtle phenotype, germlines of upshifted *cyk-4(or749ts)* worms exhibit a dramatic loss of partitions in the proximal germline that is not observed in the upshifted *zen-4(or153ts)* worms. Our conclusion is supported by RNAi experiments and by analysis of embryo production, all of which are consistent with CYK-4 not requiring ZEN-4 to cellularize oocytes.

These results raise the question of why CYK-4 can function independently of ZEN-4 in the germline, in contrast to cytokinesis where ZEN-4 is its obligate partner. In our view, the most likely explanation relates to the differential use of microtubules as a mechanism for positioning contractility in the two contexts. The proposed roles of CYK-4 in activating RhoA and/or inactivating Rac (*Green et al., 2012*; *Loria et al., 2012*; *White and Glotzer, 2012*; *Zhang and Glotzer, 2015*), and the essential role of its lipid-binding C1 domain ([*Zhang and Glotzer, 2015*]; *Figure 5C*), suggest that centralspindlin associates with the plasma membrane/cortex to function in the germline and during cytokinesis. However, there is a significant difference in the amount of centralspindlin constitutively associated with the plasma membrane/cortex in the two contexts. In the germline, centralspindlin forms a relatively stable structure that lines the rachis surface/bridges. In contrast, during contractile ring constriction in cytokinesis its most prominent localization is to microtubules and/or the overlapping microtubule bundles of the central spindle. Weak cortical localization has been reported during cytokinesis using CYK-4::GFP expressed with an exogenous *pie-1* 3' UTR that potentially drives overexpression (*Basant et al., 2015*; *Zhang and Glotzer, 2015*). However, using functional transgenes under control of the endogenous *cyk-4* promoter and 3' UTR we have been unable to detect CYK-4 at the cortex during cytokinesis in early embryos (*not shown*). A second major difference is that during cytokinesis, targeting of centralspindlin to the cortex is proposed to

be controlled by its localization to anti-parallel microtubule bundles in the central spindle (*White and Glotzer, 2012*) and by ZEN-4 motor-based movement along and accumulation at the ends of stabilized astral microtubules (*Breznau et al., 2017*; *Foe and von Dassow, 2008*; *Nishimura and Yonemura, 2006*; *Odell and Foe, 2008*; *Vale et al., 2009*). In contrast, in the region of the germline where oocyte cellularization occurs there are no centrosomes, and hence no astral microtubules (*Mikeladze-Dvali et al., 2012*), and no central spindle-like microtubule bundles (*Figure 3G*, [*Wolke et al., 2007*]). Thus, CYK-4 may function independently of ZEN-4 in the germline because the association of CYK-4 with the cortex is negatively regulated during cytokinesis to enable its control by microtubule-based signaling which generates a requirement for ZEN-4. Alternatively, association of CYK-4 with the cortex may be positively regulated to enable microtubule-independent localization in the germline by a modification or protein-protein interaction that is absent in dividing embryos. Additional work will be needed to discriminate between these potential mechanisms.

# Materials and methods

## *C. elegans* strains

*C. elegans* strains (listed in *Table 1*) were maintained at 20°C, except for strains containing temperature-sensitive alleles, which were maintained at 16°C. Transgenes generated for this study were inserted in single copy at specific chromosomal sites using the transposon-based MosSCI method (*Frøkjaer-Jensen et al., 2008*). Depending on which Mos1 insertion site was used, transgenes were cloned into pCFJ151 (*ttTi5605* site on Chr II; Uni I *oxTi185* site on Chr I; Uni IV oxTi177 site on Chr IV), pCFJ178 (cxTi10882 site on Chr IV), or pCFJ352 (*ttTi4348* site on Chr I). Transgenes were generated by injecting a mixture of repairing plasmid containing the *Cb-unc-119* selection marker and appropriate homology arms (50 ng/μL), transposase plasmid (pCFJ601, P*eft-3::Mos1 transposase*, 50 ng/μL) and four plasmids encoding fluorescent markers for negative selection against chromosomal arrays [pMA122 (P*hsp-16.41::peel-1*, 10 ng/μL), pCFJ90 (P*myo-2::mCherry*, 2.5 ng/μL), pCFJ104 (P*myo-3::mCherry*, 5 ng/μL) and pGH8 (P*rab-3::mCherry*, 10 ng/μL)] into strains EG6429 (outcrossed from EG4322; *ttTi5605*, Chr II), EG6701 (*ttTi4348*, Chr I), EG8078 (*oxTi185*, Chr I), EG6250 (cxTi10882, Chr IV), or EG8081 (*oxTi177*, Chr IV). After 1 week, progeny of injected worms were heat-shocked at 34°C for 2 hr to induce the expression of PEEL-1 to kill extra chromosomal array containing worms (*Seidel et al., 2011*). Moving worms without fluorescent markers were identified as candidates, and transgene integration was confirmed in their progeny by PCR across the junctions on both sides of the integration site.

A CRISPR/Cas9-based method (*Dickinson et al., 2015*) was used to generate an *in-situ*-tagged *gfp::zen-4* strain. Briefly, a repairing plasmid containing a loxP-flanked self-excision cassette (SEC; containing a dominant roller marker, a heat-shock inducible Cre recombinase and a hygromycin drug resistance marker) and GFP, together surrounded by homology arms for N-terminal insertion at the *zen-4* locus (651 bp of the *zen-4* 5'UTR and 541 bp of the *zen-4* coding sequence; pOD2083, 10 ng/μL) was co-injected with a plasmid encoding the Cas9 protein modified from pDD162 by inserting a guide RNA sequence (5'- AAATGTCGTCGCGTAAACG-3', 50 ng/μL) and three plasmids encoding fluorescent markers for negative selection [pCFJ90 (P*myo-2::mCherry*, 2.5 ng/μL), pCFJ104 (P*myo-3::mCherry*, 5 ng/μL) and pGH8 (P*rab-3::mCherry*, 10 ng/μL)] into the wildtype N2 strain. After a week, roller worms without fluorescent markers were singled and successful integration of GFP-SEC was confirmed by PCR spanning the homology regions on both sides. Roller worms (*gfp-SEC::zen-4/+*) were mated with the *nT1[qIs51]* balancer to facilitate SEC removal. The L1/L2 larvae of balanced roller worms (*gfp-SEC::zen-4/nT1[qIs51]*) were heat-shocked at 34°C for 4 hr to induce Cre expression and SEC removal. Normal-moving progeny without the *nT1[qIs51]* balancer (homozygous *gfp::zen-4*) were selected. Successful SEC excision and correct GFP integration were confirmed by PCR across both sides of the GFP insertion.

## RNA interference

Single-stranded RNAs (ssRNAs) were synthesized in 50 μL T3 and T7 reactions (MEGAscript, Invitrogen, Carlsbad, CA) using cleaned DNA templates generated by PCR from N2 genomic DNA or cDNA using oligonucleotides containing T3 or T7 promoters (*Table 2*). Reactions were cleaned using the MEGAclear kit (Invitrogen, Carlsbad, CA), and the 50 μL T3 and T7 reactions were mixed with 50

**Table 1.** *C. elegans* strains used in this study.

| Strain # | Genotype | Figure |
|---|---|---|
| N2 | wild type (ancestral) | 1S1, 1S2B, 2B, 3B, 5B, 5C, 5E, 5S1, 6B, 6S2 |
| OD95 | *unc-119(ed3) III; ltIs37 [pAA64; Ppie-1::mCherry::his-58; unc-119 (+)] IV; ltIs38 [pAA1; Ppie-1::GFP::PH(PLC1delta1); unc-119(+)] III* | 1S2, 2A, 2B, 3A, 3E, 3F, 3S1, 4S2, 6B |
| OD239 | *cyk-4(or749ts) III; ltIs38 [pAA1; Ppie-1::GFP::PH(PLC1delta1) unc-119 (+)] III; ltIs37 [pAA64; Ppie-1::mCherry::H2B his-58; unc-119 (+)] IV* | 2A, 2B, 3S1, 4B, 4S2 |
| OD241 | *cyk-4(t1689ts) III; ltIs38 [pAA1; Ppie-1::GFP::PH(PLC1delta1) unc-119 (+)] III; ltIs37 [pAA64; Ppie-1::mCherry::H2B his-58; unc-119 (+)] IV* | 3E, 3F, 3S1, 4B |
| OD1176 | *unc-119(ed3) III; ltSi346 [pKL3; Pcyk-4::CYK-4reencoded::mCherry; cb-unc-119(+)] IV* | 5D, 5E, 5S1D, 6C, 7C |
| OD1178 | *unc-119(ed3) III; ltSi348 [pKL4; Pcyk-4::CYK-4reencoded(R459A)::mCherry; cb-unc-119(+)] IV* | 5D, 5E, 7C |
| OD1211 | *ltSi346 [pKL3; Pcyk-4::CYK-4reencoded::mCherry; cb-unc-119(+)] IV; ltIs38 [pAA1; Ppie-1::GFP::PH(PLC1delta1); unc-119 (+)] III* | 6C |
| OD1364 | *unc-119(ed3) III; ltSi432[pKL33; Pcyk-4::CYK-4reencoded(ΔC1)::mCherry; cb-unc-119(+)] IV* | 5D, 5E |
| OD1970 | *ltSi835 [pKL62; Pcyk-4::CYK-4reencoded; cb-unc-119(+)]III; unc-119(ed3) III* | 5B, 5S1B, 5S1C, 6S2 |
| OD1972 | *ltSi837 [pKL64; Pcyk-4::CYK-4reencoded(R459A); cb-unc-119(+)] II; unc-119(ed3) III* | 5B, 5S1B, 5S1C, 6S2 |
| OD1974 | *ltSi839 [pKL65; Pcyk-4::CYK-4reencoded(R459A/K495A/R499E); cb-unc-119(+)] II; unc-119(ed3) III* | 5B, 5S1B, 5S1C, 6S2 |
| OD1978 | *ltSi843 [pKL67; Pcyk-4::CYK-4reencoded(ΔC1); cb-unc-119(+)] II; unc-119(ed3) III* | 5B, 5S1B, 5S1C |
| OD2064 | *ltSi849[pKL120; Pmex-5::mCherry-PH::tbb-2 3'UTR; cb-unc-119(+)]I; ltSi641[pKL89; Pcyk-4::CYK-4reencoded::GFP::MEI-1 (1–224); cb-unc-119(+)]I; unc-119(ed3)III; ltIs37[pAA64; pie-1/mCherry::H2B his-58; unc-119(+)] IV* | 5C |
| OD2083 | *ltSi849 [pKL120; Pmex-5::mCherry::PH(PLC1delta1)::tbb-2 3'UTR; cb-unc-119(+)] I; ltSi641 [pKL89; Pcyk-4::CYK-4reencoded::GFP::MEI-1 (1–224); cb-unc-119(+)] I; ltSi835 [pKL62; Pcyk-4::CYK-4reencoded; cb-unc-119(+)]III; unc-119(ed3) III; ltIs37 [pAA64; Ppie-1::mCherry::H2B his-58; unc-119(+)] IV* | 5C |
| OD2084 | *ltSi849 [pKL120; Pmex-5::mCherry:: PH(PLC1delta1)::tbb-2 3'UTR; cb-unc-119(+)] I; ltSi641 [pKL89; Pcyk-4::CYK-4reencoded::GFP::MEI-1 (1–224); cb-unc-119(+)] I; ltSi837 [pKL64; Pcyk-4::CYK-4reencoded(R459A); cb-unc-119(+)] II; unc-119(ed3) III; ltIs37 [pAA64; Ppie-1::mCherry::H2B his-58; unc-119(+)] IV* | 5C |
| OD2085 | *ltSi849 [pKL120; Pmex-5::mCherry::PH(PLC1delta1)::tbb-2 3'UTR; cb-unc-119(+)] I; ltSi641 [pKL89; Pcyk-4::CYK-4reencoded::GFP::MEI-1 (1–224); cb-unc-119(+)] I; ltSi839 [pKL65; Pcyk-4::CYK-4reencoded(R459A/K495A/R499E); cb-unc-119(+)] II; unc-119(ed3) III; ltIs37 [pAA64; Ppie-1::mCherry::H2B his-58; unc-119(+)] IV* | 5C |
| OD2087 | *ltSi849 [pKL120; Pmex-5::PH(PLC1delta1)::tbb-2 3'UTR; cb-unc-119(+)]I; ltSi641 [pKL89; Pcyk-4::CYK-4reencoded::GFP::MEI-1 (1–224); cb-unc-119(+)] I; ltSi843 [pKL67; Pcyk-4::CYK-4reencoded(ΔC1); cb-unc-119(+)] II; unc-119(ed3) III; ltIs37 [pAA64; pie-1::mCherry::H2B his-58; unc-119(+)] IV* | 5C |
| OD2127 | *ltSi220 [pOD1249/pSW077; Pmex-5::GFP-tbb-2-operon-linker-mCherry-his-11; cb-unc-119(+)] I; ltSi849 [pKL120; Pmex-5::mCh-PH::tbb-2 3'UTR; cb-unc-119(+)] I* | 3G |
| OD2286 | *unc-119(ed3) III; ltSi867 [pKL142; Pcyk-4::CYK-4reencoded(R459A/K495A/R499E)::mCherry; cb-unc-119(+)] IV* | 5D, 5E |
| OD3639 | *ltSi849 [pKL120; Pmex-5::mCherry::PH(PLC1delta1)::tbb-2 3'UTR; cb-unc-119(+)] I; ltSi17 [pOD928/EZ-36; Prga-3::GFP::RGA-3; cb-unc-119(+)] II; unc-119(ed3) III* | 7A, 7B |
| OD3640 | *ltSi849 [pKL120; Pmex-5::mCherry::PH(PLC1delta1)::tbb-2 3'UTR; cb-unc-119(+)] I; unc-119(ed3) III(?); zen-4(lt30[GFP::loxP::zen-4]) IV* | 1B, 3C |
| OD3686 | *ltSi849 [pKL120; Pmex-5::mCherry::PH(PLC1delta1)::tbb-2 3'UTR; cb-unc-119(+)] I; ltSi1124[pKL177/pSG092; Pcyk-4::CYK-4reencoded::mNeonGreen::cyk-4 3'-UTR; cb- unc-119(+)] II; unc-119(ed3) III* | 1B, 1C, 3D, 6A, 6D, 6S1 |
| JCC754 | *unc-119(ed3) III?; ltIs38 [pAA1; Ppie-1::GFP::PH(PLC1delta1); unc-119 (+)] III; zen-4(or153ts)IV; ltIs37 [pAA64; Ppie-1::mCherry::his-58; unc-119 (+)] IV* | 3E, 3F, 3S1 |

DOI: https://doi.org/10.7554/eLife.36919.034

**Table 2.** Oligos used for dsRNA production.

| Gene | Oligonucleotide 1 (5'→3') | Oligonucleotide 2 (5→3') | Template | Concentration (mg/ml) |
|---|---|---|---|---|
| cyk-4 (K08E3.6) | CGTAATACGACTCACTATAGGTGTCA AAGACACTCAGAAAC | CGTAATACGACTCACTATAGGCCTC TTCGAATTGGCAGCAGC | N2 cDNA | 2.0 |
| zen-4 (M03D4.1) | AATTAACCCTCACTAAAGGAATTGGT TATGGCTCCGAGA | TAATACGACTCACTATAGGATTGGA GCTGTTGGATGAGC | N2 cDNA | 1.3 |
| ect-2 (T19E10.1) | TAATACGACTCACTATAGGTCTCCGA TAAATCTGTGGGG | AATTAACCCTCACTAAAGGCAGCAG TTTGCGAAAATGAA | N2 genomic DNA | 2.0 |
| spd-1 (Y34D9A.4) | TAATACGACTCACTATAGGTCGTTGA CGCGTACTCAACT | AATTAACCCTCACTAAAGGGAATTC GAAATCCGACTCCA | N2 cDNA | 1.8 |
| rga-3/4 (K09H11.3/Y75B7AL.4) | TAATACGACTCACTATAGGCCTTCCT GAGCACGACTTTC | AATTAACCCTCACTAAAGGAGCTTT CGCGACCTTAAACA | N2 genomic DNA | 2.6 |
| rho-1 (Y51H4A.3) | AATTAACCCTCACTAAAGGATCGTC TGCGTCCACTCTCT | TAATACGACTCACTATAGGCTCGGC TGAAATTTCCAAAA | N2 genomic DNA | 1.9 |
| ced-10 (C09G12.8) | AATTAACCCTCACTAAAGGATCGCC TCATCGA AAACTTG | TAATACGACTCACTAT AGGTCAAAT GTGTCGT CGTTGGT | N2 cDNA | 2.0 |
| cdc-42 (R07G3.1) | AATTAACCCTCACTAAAGGGTTTGG CATTTTTCAGGGAA | TAATACGACTCACTATAGGACGTGT GCGTGCACATTTAT | N2 genomic DNA | 2.0 |
| hyls-1 (C05C8.9) | AATTAACCCTCACTAAAGGTGGCA AATTTTACCACTGAAA | TAATACGACTCACTATAGGTGATATC TTGTGACCGGATCA | N2 cDNA | 2.0 |
| gfp | AATTAACCCTCACTAAAGGCCAA CACTTGTCACTACTTTCTGTTATGG | TAATACGACTCACTATAGGGTATAGT TCATCCATGCCATGTGTAATCCC | Plasmid | 2.0 |

DOI: https://doi.org/10.7554/eLife.36919.035

μL of 3x soaking buffer (32.7 mM $Na_2HPO_4$, 16.5 mM $KH_2PO_4$, 6.3 mM NaCl, 14.1 mM $NH_4Cl$) and annealed (68°C for 10 min followed by 37°C for 30 min). L4 hermaphrodites were injected with dsRNA and incubated at 16 or 20°C depending on the experiment. For double or triple depletions, dsRNAs were mixed at equal concentrations (~2 μg/μl for each dsRNA); a dsRNA targeting hyls-1, a gene with no known function in the germline or early embryo, was used as a mixing control. To count number of embryos laid after RNAi-mediated depletion, L4 hermaphrodites were injected with dsRNAs and incubated at 20°C for 24 hr (or as indicated in specific experiments). Worms were singled and allowed to lay embryos at 20°C for 24 hr (or as indicated in specific experiments), adult worms were removed and all embryos and hatchlings were counted. For the imaging experiment in *Figure 5C*, the strains used also contain a cyk-4-RNAi-resistant transgene encoding CYK-4::GFP:: MEI-1 (1–224) that was originally engineered for experiments not presented in this study. L4 hermaphrodites were co-injected with cyk-4 dsRNA (to deplete endogenous CYK-4) and gfp dsRNA (to deplete CYK-4::GFP::MEI-1 (1–224)) and incubated at 20°C for 48 hr before imaging.

## Live imaging and photobleaching experiments

Germline imaging was performed by anesthetizing worms in 1 mg/ml Tricane (ethyl 3-aminobenzoate methanesulfonate salt) and 0.1 mg/ml of tetramisole hydrochloride (TMHC) dissolved in M9 for 15–30 min. Anesthetized worms were transferred to a 2% agarose pad, overlaid with coverslip, and imaged using a spinning disk confocal system (Andor Revolution XD Confocal System; Andor Technology) with a confocal scanner unit (CSU-10; Yokogawa) mounted on an inverted microscope (TE2000-E; Nikon) equipped with a 60×/1.4 Plan-Apochromat objective, solid-state 100 mW lasers, and an electron multiplication back-thinned charge-coupled device camera (iXon; Andor Technology). Germline imaging was performed by acquiring a 40 × 1 μm z-series, with no binning (*Figures 2A–B*, *3A, C, E*, *4B, 5C–D*, *6C–D* and *7A*, and *Figure 1—figure supplement 2A* and *Figure 3—figure supplement 1B*), or with 2 × 2 binning (*Figure 6B*). For *Figure 1B* a 61 × 0.5 μm (L1–L4 germlines) or 81 × 0.5 μm z-series (adult germlines), for *Figure 3D* a 129 × 0.25 μm z-series, for *Figure 3G* a 200 × 0.2 μm z-series, and for *Figure 6—figure supplement 1* an 80 × 0.5 μm z-series, all with no binning, were collected.

For the rachis bridge diameter measurement in *Figure 1C*, young adult germlines were imaged by acquiring a 129 × 0.25 μm z-series with no binning, then cross sections were used to measure the length of the CYK-4::mNeonGreen signal across the widest opening of each rachis bridge. For the

rachis bridge diameter measurement in *Figure 4—figure supplement 2*, adult germlines were imaged by acquiring a 40 × 1 µm z-series with no binning, then cross sections were used to measure the length of the GFP::PH signal across the widest opening of each rachis bridge.

The photobleaching experiments in *Figure 6A* (CYK-4::mNeonGreen and mCherry::PH) and *Figure 7B* (GFP::RGA-3 and mCherry::PH) were performed on a Zeiss LSM 880 confocal microscope using a 63×/1.4 oil Plan-Apochromat objective. Single z-plane images were acquired at 30 s intervals. After acquisition of the first two cycles of images, a region covering the germline rachis was photobleached, followed by acquisition of at least 12 more cycles.

## Construction of worm trap microfluidic devices

The worm trap microfluidic device (*Figure 4—figure supplement 1*) is assembled out of a 7-mm-thick polydimethylsiloxane (PDMS) chip with microchannels engraved on its surface and a 35 × 50 mm #1.5 microscope coverslip, which seals the microchannels. The microchannels of the device are of three different depths, 10, 50, and 750 µm. PDMS chips were cast out of a 50/50 mixture of Sylgard 184 (Dow-Corning) and XP-592 (Silicones Inc.) silicone pre-polymers, each of which was a 10:1 mixture of the base and cross-linker components of the respective silicone material. The master mold used to cast the chips was a 5-inch silicon wafer with a photolithographically fabricated microstructure with 10, 50, and 750 µm tall features made of UV-cross-linked photoresists of the SU8 family (MicroChem, Newton, MA). To fabricate the master mold, the wafer was spin-coated with SU8 2005 to a thickness of 10 µm, baked and exposed to UV light through a photomask. After that, the wafer was spin-coated with SU8 2015 to a cumulative thickness of 50 µm, exposed through another photomask, and baked. Finally, the wafer was spin-coated with SU8 2150 to a cumulative thickness of 750 µm, exposed through a third photomask, baked, and developed.

To facilitate the immobilization of worms in the imaging chambers, the surfaces of both PDMS (ceiling of the imaging chambers) and glass coverslip (floor of the imaging chambers) were roughened by coating them with 0.2 µm polystyrene beads, which were covalently attached to the surfaces using carboxyl-amine chemistry. (The size of the beads was too small to cause substantial scattering of light, making the beads practically invisible under a microscope.) To this end, the surfaces of both the coverslip and PDMS chip were oxidized by exposing them for 10 s to oxygen plasma (using a PlasmaPreen II plasma treater by Plastmatic Systems Inc.) and immediately treated with a solution aminopropyltrimethoxysilane (4% in EtOH) and then incubated under this solution for 5 min at room temperature (in a fume hood). As a result of this treatment, the glass and PDMS surfaces were coated with amine groups. The coverslip and PDMS chip were rinsed with ethanol, dried with filtered compressed air, and then incubated for 30 min at room temperature under a 0.01% suspension of 0.20 µm carboxylated polystyrene beads (Polybead Carboxylate Microspheres, from Polysciences) in a 20 mM solution of HEPES, pH 8, containing 0.01% EDC [1-Ethyl-3-(3-dimethylaminopropyl)carbodiimide; to promote the reaction between carboxyl groups on the beads and amine groups on the surface of glass or PDMS]. The coverslip and PDMS chip were then rinsed with water and dried.

## Experiments in worm trap microfluidic device

To load *cyk-4* mutant animals (OD239 (*cyk-4 (or749ts)*) and OD241 (*cyk-4(t1689ts)*)) into the worm trap device, the PDMS chip was placed with its microchannels facing up, and a 1 ul droplet of media (0.5xPBS + 1:10 dilution of HB101 overnight culture) was dispensed onto the chip near the center of each of the three 50 µm deep, 3.7 × 3 mm rectangular imaging chambers on the chip surface (the drops remained separate because of hydrophobicity of PDMS). One or more young adult worms were carefully transferred into each drop using a platinum pick. The chip was inverted, and the engraved bottom side was brought in contact with the coverslip; this turned the microgrooves into sealed microchannels and the drop medium filled these microchannels. To hold the chip and coverslip together, a deep O-shaped channel, which surrounded the liquid-filled microchannels of the device and served as a vacuum cup, was connected to a regulated source of vacuum that was set at a low level of −5 kPa in the gauge pressure (low vacuum). The application of vacuum resulted in a partial collapse of the imaging chambers, reducing their depth to ~45 µm, which did not prevent worms from moving freely and feeding. To trap worms for high-resolution imaging, the adjustable vacuum level was dialed up to −25 kPa (high vacuum), causing a major reduction of the imaging

chamber depths and immediate immobilization of worms in the chambers. If immobilized worms were poorly positioned, the vacuum was dialed back to low, allowing worms to move, and then to high again, re-trapping worms for imaging. The immobilization was most effective away from the edges of the imaging chambers. Imaging was performed on a spinning disk confocal microscope (Andor Revolution XD Confocal System; Andor Technology) as described above. A stack of 40 images with a 1 μm step along the Z-axis was collected. After imaging was completed, the vacuum was switched from high to low, allowing worms to move, feed, and recover until next imaging timepoint.

Initial stacks of confocal images in the temperature shift experiments were acquired immediately after the device was assembled, without perfusion of a bacterial culture through the device. At temperature shift, the device inlet was connected through a tubing line to a reservoir with bacterial culture (which was immersed in a 38°C water bath), the device outlet was connected to a reservoir with plain medium, and a continuous perfusion of bacterial culture through the imaging chambers was initiated by setting a constant positive differential pressure between the inlet and outlet (by lifting the inlet reservoir above the outlet reservoir). In preparation for temperature shift to 25°C, a warm air blower was positioned adjacent to the stage, with its air flow directed toward the worm trap device and the tubing line connected to the device inlet. A temperature probe was affixed to the stage adjacent to the device to monitor the temperature throughout the course of the experiment; the blower was manually adjusted to maintain air temperature on the stage at 25.5–26.5°C during the course of the experiment.

Prior to trapping and imaging at the initial restrictive time point, worms were exposed to the restrictive temperature for ~45 min. Subsequently, worms were trapped and imaged every 45–60 min over a period of 5 hr. For most experiments, two worm trap devices were used in parallel to enable side-by-side analysis of the two *cyk-4* mutants (*t1689ts,* and *or749ts*). Each device was operated independently of the other with respect to both perfusion and the application of vacuum. (Two separate vacuum regulators set at −5 and −25 kPa made it possible to independently apply either low or high vacuum to either device.) Thus, initial perfusion, trapping and imaging was offset by 20–30 min and subsequent imaging alternated between each of the two traps every 20–30 min during the time course. Three independent experiments were performed for each strain (n = 7 for *cyk-4 (t1689ts),* and n = 9 for *cyk-4(or749ts)*).

## Image analysis

All images were processed, scaled, and analyzed using ImageJ (National Institutes of Health) or MetaMorph software (Molecular Devices).

## Western blotting

To reduce the contamination with *E. coli* (which the worms eat), 45 worms from each of the indicated conditions were transferred onto a medium plate with no food and the plate was flooded with 4 ml of M9. After 1 hr, when the *E. coli* have largely been flushed out of the worms' digestive tracts, the worms were transferred into a tube containing 500 μl of M9 in a screw-cap 1.5 ml tube, 0.5 ml of M9 + 0.1% Triton X-100 was added, and worms were pelleted by centrifuging at 400 x g for 1–2 min. Buffer was removed leaving behind 50 μl and worms were washed and pelleted three more times in 1 ml of M9 + 0.1% Triton X-100. After the last wash, buffer was removed leaving behind 30 μl and 10 μl of 4X sample buffer was added. Samples were placed in a sonicating water bath at 70°C for 10 min, boiled in a 95°C heating block for 5 min, and sonicated again for 10 min at 70°C before freezing. Samples were thawed and separated by SDS-PAGE. After transferring the separated protein bands to a nitrocellulose membrane, the blot was cut into two parts at ~70 kDa. The above-70-kDa blot was probed using 1 μg/ml of rabbit anti-CYK-4 (aa 131–345), which was detected using an HRP-conjugated secondary antibody (1:10,000; GE Healthcare Life Sciences) and WesternBright Sirius detection system (Advansta). The below-70-kDa blot was probed for α-tubulin using the monoclonal DM1α antibody (1:500; Sigma-Aldrich), and then visualized by Western Blue Stabilized Substrate for Alkaline Phosphatase (1:2000; Promega). Antibodies against CYK-4 were generated by injecting a purified GST fusion with CYK-4 aa 131–345 into rabbits (Covance). Antibodies were purified from serum using standard procedures (*Harlow and Lane, 1988*) on a 1 ml NHS HiTrap column (GE Healthcare Life Sciences) containing an immobilized MBP fusion with the same region of CYK-4.

## Positional correlative anatomy EM

Samples were high-pressure frozen followed by freeze-substitution as described previously (*Kolotuev et al., 2009*) and flat embedded, targeted and sectioned using the positional correlation and tight trimming approach (*Kolotuev, 2014*). For ultramicrotome sectioning, a Leica UC7 microtome was used. 100 nm sections were collected on the slot-formvar coated grids and observed using a JEOL JEM 1400 TEM microscope (JEOL, Japan). Samples were aligned and rendered using the ImageJ and IMOD programs.

## Acknowledgements

We thank the electron microscopy facilities in Rennes and Lausanne, and Tom Garber for assistance in constructing the EM model. JSG-C was supported by the University of California, San Diego Cancer Cell Biology Training Program (T32 CA067754). AD and KO receive salary and other support from the Ludwig Institute for Cancer Research. AG and EG were supported by an award from the National Science Foundation (PHY-14113130 to AG).

## Additional information

### Funding

| Funder | Grant reference number | Author |
| --- | --- | --- |
| National Institutes of Health | T32 CA067754 | J Sebastian Gomez-Cavazos |
| National Science Foundation | PHY-14113130 | Alex Groisman |
| Ludwig Institute for Cancer Research | | Arshad Desai Karen Oegema |

The funders had no role in study design, data collection and interpretation, or the decision to submit the work for publication.

### Author contributions

Kian-Yong Lee, Conceptualization, Investigation, Writing—original draft, Writing—review and editing; Rebecca A Green, Conceptualization, Investigation, Methodology, Writing—original draft, Writing—review and editing; Edgar Gutierrez, Conceptualization, Investigation, Methodology; J Sebastian Gomez-Cavazos, Conceptualization, Investigation; Irina Kolotuev, Conceptualization, Investigation, Methodology, Performed the transmission electron microscopy after fixation by high pressure freezing and freeze-substitution shown in Figure 1-figure supplement 2; Shaohe Wang, Investigation, Generated the in situ-tagged GFP::ZEN-4 C. elegans strain that was used in this project; Arshad Desai, Conceptualization, Supervision, Writing—original draft, Writing—review and editing; Alex Groisman, Karen Oegema, Conceptualization, Resources, Supervision, Funding acquisition, Writing—original draft, Project administration, Writing—review and editing

### Author ORCIDs

Kian-Yong Lee http://orcid.org/0000-0003-0946-1703
Irina Kolotuev https://orcid.org/0000-0003-1433-8048
Arshad Desai http://orcid.org/0000-0002-5410-1830
Karen Oegema http://orcid.org/0000-0001-8515-7514

### Decision letter and Author response

Decision letter https://doi.org/10.7554/eLife.36919.038
Author response https://doi.org/10.7554/eLife.36919.039

## Additional files

### Supplementary files

- Transparent reporting form

DOI: https://doi.org/10.7554/eLife.36919.036

**Data availability**

All data generated or analyzed during this study are included in the manuscript and supporting files.

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
