## [Decision Letter]

[Editors’ note: a previous version of this study was rejected after peer review, but the authors submitted for reconsideration. The first decision letter after peer review is shown below.]

Thank you for submitting your work entitled "CYK-4 functions independently of its centralspindlin partner ZEN-4 to cellularize oocytes in germline syncytia" for consideration by *eLife*. Your article has been reviewed by two peer reviewers, and the evaluation has been overseen by a Reviewing Editor and a Senior Editor. The reviewers have opted to remain anonymous.

Our decision has been reached after consultation between the reviewers. Both referees felt that your work showing that cellularization of oocytes is dependent on *cyk4*, yet, independent of *zen4* is interesting. However, the referees concurred that the story is not mechanistically advanced enough to provide insight into the molecular role of Cyk4 in this interesting cellularization process.

In discussions, both referees felt that the required effort to address the raised concerns would take over two months and therefore we will be unable to consider the manuscript further presently, as per *eLife* revision policy.

However, if you are able to provide further molecular insight into the role of *cyk4* in cellularization in the germline, based on experiments suggested by the referees, we will be willing consider a reworked manuscript as a new submission. I will send it back to the same referees, in this case. The experiments largely are aimed to address the relationship between Rho-GTPase and Cyk4 and to identify the execution point of Cyk4 in cellularization.

The detailed comments of Reviewer #1 and 2 are pasted below.

*Reviewer #1:*

The main finding of the paper is that CYK-4 is required for oocyte cellularization in *C. elegans*. CYK-4 function in this process appears to be independent of ZEN-4. Furthermore, the authors identified two domains in the C-terminus of CYK-4, its C1and GAP domains, which are required for its localization to the rachis surface and its function in cellularization. Finally, they show that CYK-4 contributes to the level of active RhoA (deduced by RGA-3 levels) at the rachis bridge.

The question of how oocytes are cellularized in a syncytial germline is of great interest. This paper identifies CYK-4 as an important player, but it falls short on explaining the molecular mechanism through which CYK-4 acts. The claim that ZEN-4 is not involved is curious, but doesn't contribute to our understanding of the process at hand. The potential involvement of RhoA provides a hint of mechanistic insight. However, this angle is not explored deeply and is not substantiated, as explained in the following comments:

1) How is the localization of CYK-4 affected by partial Rho-1 depletion? The authors claim that they couldn't examine CYK-4 localization in Rho-1 RNAi because the gonad was severely disrupted. However, in Figure 5D middle panel they show a partial RHO-1 depletion that shows absence of RGA-3 from the rachis bridge. Where is CYK-4 in this case?

2) The authors claim a feedback loop in which RHO-1 recruits CYK-4 and CYK-4 increases RHO-1 activity. Is it active RHO-1 that is required for the recruitment of CYK-4? Instead of depleting RHO-1 the authors could deplete the RhoGEF that activates RHO-1 and see if this affects CYK-4 recruitment.

3) How does CYK-4, which is a RhoGAP, contribute toward the activation of RHO-1? If the authors envision a mechanism analogous to what the Glotzer lab showed for cytokinesis, then defects resulting from loss of CYK-4 should be rescued by activating mutations in ECT-2 or depletion of RGA-3/4.

4) Is the activation of RHO-1 by CYK-4 at the rachis essential for cellularization? How exactly does CYK-4 contribute to cellularization?

In addition, the paper's claim that ZEN-4 has no apparent function in the germline suffers from the following caveats:

1) A previous study has shown that same allele of ZEN-4 *zen-4(or153ts)* had severe defects in the germline with multinucleated germ cells (Zhou et al., 2013). The authors should address the discrepancy between their negative findings and the published germline defects.

2) The authors argue that ZEN-4 is not required for compartment bridge closure in the germline because they did not observe any microtubule bundles passing through the compartment bridges in the pachytene, loop, or oocyte cellularization regions of the germline. However, I do see some filament-like structures in their figures, despite the low resolution. Moreover, a previous study (Wolke et al., 2007) has shown with an α-tubulin staining the presence of microtubules running from the rachis into the growing oocytes.

*Reviewer #2:*

In this manuscript, the authors examined the role of centralspindlin in the intercellular bridges of the *C. elegans* syncytial germ line. Although both subunits of centralspindlin localize to the intercellular bridges between germ cells and the rachis (or rachis surface more broadly?), ZEN-4 was dispensable for oocyte formation and only CYK-4 was required. By comparing the phenotypes of various *cyk-4* and *zen-4* mutants defective for the interaction between them and for the functions of CYK-4 C1 and GAP domains, they concluded that the C1 and GAP domains are important for oocyte cellularization, CYK-4 recruitment to the rachis surface and enrichment of active RhoA. Most of the experiments were carefully designed and performed. However, it remains unclear how CYK-4 contributes to the maintenance of germline structure and cellularization of oocytes as detailed below. In addition, some of the summaries of previous works are imprecise.

1) Localization of CYK-4

Although the localization pattern of CYK-4 in the gonad is described "to bridges" in the Abstract, "rachis surface" is used in most places in the main text. Actually, CYK-4::mNeonGreen seems to be enriched at the peripheries of the ring-like openings (Figure 1B and Figure 1—figure supplement 3). Discrimination between flat localization on the rachis surface (plasma membrane or cell cortex) and accumulation at the edges of the ring-like openings is important. For example, in the images presented in Figure 2D, accumulation at the ring-like openings seems to be weaker in *zen-4(RNAi)* than in the control. When CYK-4 localization is examined, images comparable to those in Figure 1—figure supplement 3 should be presented.

2) Compartmentalization and incomplete cytokinesis

In contrast to the formation of the intercellular connection between the primordial germ cells (Z2 and Z3), it is not clear whether and how compartmentalization of the newly generated gem cells is coupled with mitosis. Is the compartmentalization at the distal tip of adult gonad also incomplete cytokinesis?

3) Point of CYK-4 function during oogenesis

Although the authors emphasize the role of CYK-4 in the intercellular bridge closure, this is not very strongly supported by the data presented. After compartmentalization at the distal tip, the openings of the bridge have to be stably maintained for a long time until the nucleus reaches the proximal tip and completes cellularization. Although the authors performed longitudinal live observation of the temperature-sensitive *cyk-4* mutants in Figure 3, interpretation of the result is not straight-forward. First, even at time 0 of temperature upshift, the gonad of the *cyk-4(or769)* mutant looks different from that of the wild-type. Widening of the bridge openings is observed at the transition zone between pachytene and loop regions. In the mutant adult gonad in Figure 1D, multinucleation is observed in the distal cells. These suggest that CYK-4 plays a role in maintenance of the circular openings of the compartment bridges. To clearly discriminate oocyte cellularization from the maintenance of the bridges, normal maintenance of the bridge diameter until the moment of cellularization need to be confirmed by more detailed analyses such as Figure 1—figure supplement 3.

4) Mechanism of CYK-4 recruitment to the rachis surface

The model in Figure 6 implies the role of interaction with RHO-1 for the recruitment of CYK-4 to the rachis surface. Moreover, a positive feedback loop is mentioned (subsection “Active RhoA localizes to the rachis surface and collaborates with CYK-4 to maintain germline structure”, last paragraph). However, the effect of RHO-1 depletion on CYK-4 localization was not shown. Although the authors claim that penetrant *rho-1(RNAi)* disrupts the rachis surface, partial *rho-1(RNAi)* equivalent to Figure 5D should be possible. Alternatively, acute inhibition of RHO-1 can be achieved by injection of C3 enzyme?

It is counter-intuitive that a GAP promotes GTP-form of RHO-1. However, activation of RHO-1 by CYK-4 through positive regulation of ECT-2 GEF has been reported in the context of embryonic cytokinesis (Zhang and Glotzer 2015). Does ECT-2 play a role also in the gonadal function of CYK-4?

5) Functionality the FP-tagged *cyk-4* transgenes

Although this is not crucial for the authors' conclusion, both the *cyk-4::mNG* and *cyk-4::mCh* strains showed 6 and 11% embryonic lethality after *cyk-4(RNAi)* treatment. Is this because these transgenes are not 100% functional or because RNAi-resistance is not perfect? Or, does dsRNA injection affect the quality of the embryos? Has a transgene without FP-tagging been tried?

6) Worm trap microfluidic chip

The shapes of the devices in Figure 3 and Figure 3—figure supplement 1 look different. The outline of the device should be illustrated in the supplement.

7) Introduction, fourth paragraph

Although Plk1-phosphorylation sites reside in the N-terminal half of human Cyk4/MgcRacGAP1, they are not at the N-terminus of Cyk4. They are in the linker between the coiled coil and the C1 domain.

8) "During this transition, centralspindlin is released from microtubules and concentrates in the midbody ring (Elia, Sougrat, Spurlin, Hurley, and Lippincott-Schwartz, 2011; Green et al., 2013; Hu, Coughlin, and Mitchison, 2012)".

This is incorrect. GFP-MKLP1 appears as a disk at the midbodies (Elia 2011 and Hu 2012). The ring-like appearance of centralspindlin by immunofluorescence is known to be an artifact due to the poor accessibility of the antibodies into the central core of the midbody matrix. Although Hu 2012 described the IF image of MKLP1 as "centralspindlin remained at the center and appeared as a single ring in some images" (Hu 2012, Figure 1C), side by side comparison of GFP fusion and IF staining clearly demonstrated that GFP-MKLP1 filled the dark zone detected by IF (Hu 2012, Figure 2D). Furthermore, based on the super-resolution observation of GFP-MKLP1 by structured illumination, Elia 2011 explicitly describes it as "disk-like distribution". No single image of a clear ring-like midbody localization of centralspindlin is found in Green 2013. Finally, immuno-EM with an anti-MKLP1 antibody also supports the disk-like localization (Elad et al., (2010). Microtubule organization in the final stages of cytokinesis as revealed by cryo-electron tomography. J Cell Sci 124, 207-215). While it is possible to speculate that there might be a short time window in which centralspindlin is relocalized from a disk to a ring, no convincing data such as time-lapse super-resolution observation have been published.

[Editors’ note: what now follows is the decision letter after the authors submitted for further consideration.]

Thank you for resubmitting your work entitled "CYK-4 functions independently of its centralspindlin partner ZEN-4 to cellularize oocytes in germline syncytia" for further consideration at *eLife*. Your revised article has been favorably evaluated by Anna Akhmanova (Senior Editor), a Reviewing Editor, and two reviewers.

The manuscript has been improved but there are some remaining issues that need to be addressed before acceptance, as outlined below:

There is one piece of data that both referees would like included, since this was a question that arose during the original reviews and has been raised again. This has to do with the how *cyk-4* activates *rho-1* and what role does *ect-2* play in this, and seems like you already have this piece of data. My suggestion would be to include data that suggests that the mechanism is different from the Glotzer lab mechanism. You can then publish the exact mechanism in a later study.

Other points that I would like you to consider and provide explanations and/or alter text accordingly.

1) Possible role of CYK-4 in the establishment/maintenance of the channels that are narrow enough

I appreciate that the authors examined the influence of the *cyk-4* GAP mutant on the diameter of the rachis bridges (Figure 4—figure supplement 2). Their data indicate that the bridges are open more widely in the *cyk-4* mutant. This raises a possibility that the failure of cellularization could be an indirect consequence of a failure in establishment or maintenance of the channels to be narrow enough for successful cellularization. For example, one can imagine a scenario that cellarization itself is achieved without CYK-4 but can't work if the opening is too wide.

2) Ring-like localization of centralspindlin in mammalian cells

The point in the previous comment was that the ring-like localization of centralspindlin in mammalian cultured cells has not been established. The arguments in the rebuttal are completely out of focus. In Hu 2012, the ring-like pattern was shown only by IF but not by live GFP-MKLP1 imaging. In Elia 2011, which employed super-resolution microscopy, the localization of MKLP1 was explicitly described as "disk-like". Immuno-EM in Elad 2010, which has been neglected by the authors, confirmed uniform localization of MKLP1 epitopes across the midbody. In their rebuttal, the authors didn't argue against these points. Nonetheless, in the current manuscript they still describe "These results suggest that ZEN-4 can target to the intercellular bridge independently of midbody microtubules, which is reminiscent of work in cultured human cells describing localization of centralspindlin to a ring-like structure during the late stages of cytokinesis (Elia et al., 2011; Hu et al., 2012)." The clause, "which is reminiscent…" should be removed since this is imprecise and misleading.

This is the point that requires some experimental data.

3) How does CYK-4, which is a RhoGAP, contribute toward the activation of RHO-1? If the authors envision a mechanism analogous to what the Glotzer lab showed for cytokinesis, then defects resulting from loss of CYK-4 should be rescued by activating mutations in ECT-2 or depletion of RGA-3/4."

In response to point 3, raised in the original manuscript, the authors provided in their reply to reviewers a detailed explanation for the mechanism they propose for RHO-1 activation by CYK-4, which, importantly, is distinct from the mechanism proposed by the Glotzer lab. The evidence for their proposed mechanism is "In ongoing work from our group that will be the basis for another publication". Importantly, they did not add to the manuscript any new data regarding this point. Only in the Discussion they added the following sentence: "We suspect that the role of CYK-4 in generating active RhoA is mediated by the region between its coiled-coil and C1 domains that work in human cells has previously implicated in binding to the Ect2 GEF (Burkard et al., 2009; Wolfe et al., 2009)."

---

## [Author Response]

[Editors’ note: the author responses to the first round of peer review follow.]

Reviewer #1:The main finding of the paper is that CYK-4 is required for oocyte cellularization in C. elegans. CYK-4 function in this process appears to be independent of ZEN-4. Furthermore, the authors identified two domains in the C-terminus of CYK-4, its C1and GAP domains, which are required for its localization to the rachis surface and its function in cellularization. Finally, they show that CYK-4 contributes to the level of active RhoA (deduced by RGA-3 levels) at the rachis bridge.The question of how oocytes are cellularized in a syncytial germline is of great interest. This paper identifies CYK-4 as an important player, but it falls short on explaining the molecular mechanism through which CYK-4 acts. The claim that ZEN-4 is not involved is curious, but doesn't contribute to our understanding of the process at hand. The potential involvement of RhoA provides a hint of mechanistic insight. However, this angle is not explored deeply and is not substantiated, as explained in the following comments:1) How is the localization of CYK-4 affected by partial Rho-1 depletion? The authors claim that they couldn't examine CYK-4 localization in Rho-1 RNAi because the gonad was severely disrupted. However, in Figure 5D middle panel they show a partial RHO-1 depletion that shows absence of RGA-3 from the rachis bridge. Where is CYK-4 in this case?2) The authors claim a feedback loop in which RHO-1 recruits CYK-4 and CYK-4 increases RHO-1 activity. Is it active RHO-1 that is required for the recruitment of CYK-4? Instead of depleting RHO-1 the authors could deplete the RhoGEF that activates RHO-1 and see if this affects CYK-4 recruitment.

In our original manuscript, we showed that the Rho GTPase binding interface of the CYK-4 GAP domain is required for CYK4 to localize to the rachis surface and rachis bridges. We show that the Rho GTPase binding interface does not promote CYK-4 localization by binding to Rac or Cdc42. However, RhoA inhibition leads to a phenotype comparable to CYK-4 inhibition, suggesting that an interaction with RhoA could be required to recruit CYK-4 to the rachis surface/bridges. To understand how the Rho GTPase binding interface of the CYK-4 GAP domain contributes to CYK-4 targeting, we began by performing a photobleaching experiment to characterize CYK-4 dynamics (new panel added as Figure 6A in the revision) to determine if the CYK-4-containing structure that lines the rachis and rachis bridges is rapidly turning over. We photobleached CYK-4::mNeonGreen in a section of the rachis in the pachytene region and imaged over a 5-minute interval following the bleach. No recovery of the CYK-4::mNeonGreen was observed over this interval, suggesting that once it has been built, the CYK-4 containing structure that lines the rachis surface/bridges turns over relatively slowly.

As the reviewers highlight, localization of CYK-4 to the rachis surface and rachis bridges cannot be analyzed when RhoA^RHO-1^ or ECT-2 are fully depleted because the rachis surface/bridges are lost. We therefore performed the reviewer-suggested experiments and analyzed the amount of CYK-4 on the rachis surface in worms partially depleted of RhoA^RHO-1^ or ECT-2 (new panels added as Figure 6D in the revision). Given that the CYK-4-containing structure that lines the rachis and rachis bridges seems to be fairly stable, we expect that CYK-4 might not be completely lost under these conditions. For the experiment, worms expressing CYK-4::mNeonGreen along with an mCherry tagged plasma membrane probe were injected with dsRNA targeting RhoA^RHO-1^ or ECT-2 at the L4 stage, and their germlines were imaged 20 hours later, prior to the point when the rachis surface/bridges have completely disintegrated. This experiment revealed that CYK4 levels on the rachis surface/bridges were qualitatively reduced by both perturbations, consistent with the idea that the Rho GTPase binding interface of the CYK-4 GAP domain promotes CYK-4 localization by binding to active RhoA^RHO-1^. We note that this experiment does not distinguish between the reduction in CYK-4 levels occurring because an interaction between the CYK-4 GAP domain and RhoA^RHO-1^ is required for the initial localization of CYK-4 or for its stable maintenance. We also cannot formally rule out the possibility that the Rho GTPase binding interface of the CYK-4 GAP domain is required for CYK-4 deposition via an interaction with an as yet uncharacterized protein, and the reduction in CYK-4 levels when RhoA^RHO-1^ is depleted is an indirect consequence of the fact that RhoA^RHO-1^ depletion compromises the structure of the rachis lining/bridges. However, we think this last possibility is unlikely since the set of proteins required for germline function has been comprehensively characterized (Green et al., 2011) and did not yield an unexpected candidate whose inhibition results in phenotype analogous to the one observed following CYK-4, ECT-2 or RhoA^RHO-1^ depletion.

3) How does CYK-4, which is a RhoGAP, contribute toward the activation of RHO-1? If the authors envision a mechanism analogous to what the Glotzer lab showed for cytokinesis, then defects resulting from loss of CYK-4 should be rescued by activating mutations in ECT-2 or depletion of RGA-3/4.4) Is the activation of RHO-1 by CYK-4 at the rachis essential for cellularization? How exactly does CYK-4 contribute to cellularization?

Collectively, the reviewers ask for more information about the proposed contribution of CYK-4 to RhoA activation, and whether we envision a mechanism analogous to the model proposed by the Glotzer lab. The mechanism we propose is distinct from the mechanism proposed by the Glotzer group. Prior work in human cells identified a region in the N-terminal half of the human homolog of CYK-4 between the coiled-coil and the C1 domain that interacts with the RhoA GEF Ect2 when it is phosphorylated by Plk1. Thus, we think that the C1 and GAP domains at the CYK-4 C-terminus collaborate to recruit CYK-4 to the rachis surface/bridges, and, once there, CYK-4 interacts with and activates ECT-2 via the domain in its N-terminal half. In our model, CYK-4 requires active RhoA to be incorporated into the structure that lines the rachis surface/bridges and is, in turn, required to activate RhoA at this location (which would represent a positive feedback loop).

Based on their work in the embryo, the Glotzer lab found that mutations in the Rho GTPase-binding interface of the GAP domain reduced RhoA activation during cytokinesis (PMID: 26252513). Our work would suggest that this is likely to be due to a reduction in the ability of the CYK-4 to localize to the plasma membrane. Since the Glotzer lab did not realize that the GAP domain could have a role in targeting CYK-4 to the plasma membrane, they explain the reduction in RhoA activation in the Rho GTPase binding interface mutants by proposing that the GAP domain interacts directly with the GEF domain to activate it in a fashion that depends on the ability of the GAP domain to bind RhoA (see model in Figure 7E of their paper, PMID: 26252513). To support this idea, they show a pull down assay in a supplemental panel to show that the CYK-4 GAP domain can interact with the GEF domain (PMID: 26252513, Figure 7—figure supplement 1). They also attempted in vitro assays, but were unable to show that the GAP domain is able activate ECT-2 in vitro.

In summary, both studies suggest that CYK-4 is important for RhoA activation and that the Rho GTPase binding interface of the CYK-4 GAP domain contributes to this function. Where our models differ is in how we propose that the Rho binding interface contributes to RhoA activation. Our work in the germline suggests that the GAP domain is required to recruit CYK-4 to the plasma membrane. The Glotzer lab proposes that the GAP domain interacts with the ECT-2 GEF domain in a RhoA-dependent fashion to activate it.

The reviewer is correct in saying that the Glotzer lab showed that during cytokinesis, inhibition of RGA-3 and RGA-4, which are the main GAPs that oppose RhoA activation in the germline and early embryo, is able to rescue the furrowing defects resulting from mutations in the Rho GTPase binding interface of the GAP domain (PMID: 26252513). The rescue was quite reasonable for the R459A mutant, which presumably disrupts the interface less than their EE mutant (analogous to our AAE mutant), which was only very poorly rescued. This suggests that the Rho GTPase interface mutants (particularly the R459A mutant) retain some ability to activate RhoA during cytokinesis and this can be rendered sufficient for furrowing by removal of the main negative regulators that suppress RhoA activation. To address the reviewer’s question of whether this is also true in the germline, we tested whether RGA-3/4 depletion could rescue the effect of the R459A mutation on embryo production (included as new Figure 7C). Our results indicate that in the germline, RGA-3/4 depletion cannot rescue the effect of the R459A mutation on embryo production, consistent with our observation that the R459A mutant protein fails to localize to the rachis surface and is therefore unlikely to retain function.

The comparison of our data with the Glotzer data suggests that, in the embryo, the R459A mutant of CYK-4 may maintain some ability to localize and this can be made sufficient for furrowing by depleting the RGA-3/4 RhoGAP, but this is not true in the germline.

In addition, the paper's claim that ZEN-4 has no apparent function in the germline suffers from the following caveats:1) A previous study has shown that same allele of ZEN-4 zen-4(or153ts) had severe defects in the germline with multinucleated germ cells (Zhou et al., 2013). The authors should address the discrepancy between their negative findings and the published germline defects.

We have now included an additional supplemental figure (new Figure 3—figure supplement 1) and a paragraph to the Discussion that enable a clear comparison between our results and the prior work. This section highlights the points where the two studies agree and also a key point, identified by the Reviewer, where our findings contradict a conclusion made in the prior work.

In our paper, we perform a careful analysis of germline structure and embryo production following RNAi-mediated depletion of CYK-4 and ZEN-4 beginning at the L4 stage as well as following upshift of fast-acting temperature sensitive mutants at the L4 stage. All of our results support the conclusion that CYK-4 plays a critical role in germline structure and embryo production in the adult, whereas ZEN-4 is not essential. As noted by the reviewer, this is in contrast to a prior analysis of fixed extruded germlines after a similar upshift, in which the authors suggested that *cyk-4(t1689ts*) and *zen-4(or153ts*) germlines exhibit a similar phenotype (Zhou et al., 2013).

A direct comparison is difficult because the prior report was not specific about the nature of the defects in the upshifted *cyk-4(or749ts)* versus *zen-4(or153ts)* mutant germlines, and the phenotypes were not quantified. The Zhou et al. paper showed a single image of germlines from each mutant. Notably, while they report a defect in the proximal germline of *cyk-4(or749ts)* that resembles the defect we analyzed in depth in our manuscript, no image of a comparable defect was shown or reported for *zen-4(or153ts).* Instead, in the image in their figure, they highlight small patches of multinucleated compartments in the pachytene region of germlines from the two mutants following a 12-hour upshift at the adult stage (exactly how old the adults were is also unclear). In our live imaging-based analysis of adult worms upshifted for 28 hours beginning at the L4 stage, we have observed rare bi-nucleate compartments in the pachytene region in the *zen-4(or153ts)* strain (especially in older worms). However, these bi-nucleation events appear to be culled through apoptosis at the loop region of the germline, as we have never observed bi-nucleation in oocytes. In contrast to this infrequently observed subtle phenotype in germlines from the upshifted *zen-4(or153ts)* strain, germlines of upshifted *cyk-4(or749ts)* mutant worms exhibit a dramatic loss of partitions in the proximal region of the germline. Our analysis of embryo production confirms the difference in phenotype: upshifted *zen-4(or153ts)* worms continue to produce embryos (Figure 3Fin the revision) whereas upshifted *cyk-4(or749ts)* worms are completely sterile (Figure 2B in the revision). The Zhou et al. manuscript did not analyze embryo production following temperature upshift.

Another complication in the Zhou et al. paper is that they do two very different types of perturbation, feeding beginning at the L1 stage, which has the potential to disrupt germline development, and a temperature upshift in the adult, which would test for a specific role in embryo production. They claim that both perturbations lead to similar effects in both mutants, which our analysis indicates is not the case. To clarify the roles of the CYK-4 Cterminal C1-GAP module and the interaction between the CYK-4 and ZEN-4 dimers (centralspindlin assembly) during germline development versus during embryo production in the adult, we now report the results of two types of upshift experiments for all three mutants (expanded Figure 2, new Figure 3—figure supplement 1). To assess the effect of each temperature-sensitive mutation on worm and germline development, we monitored the effect of upshifting the worms between the L1 and L4 stages. To monitor the effect on oocyte cellularization, we assessed the effect of upshifting the mutants between the L4 and adult stages. All three mutants have been shown to be fast acting (<90s). Prior work on the mutant that perturbs the C-terminal C1-GAP module of CYK-4 (*cyk-4(or749ts)*) has shown that it retains the ability to interact with ZEN-4 and to support midzone assembly but is unable to promote contractile ring assembly (PMID: 19056985) and fails to localize to the rachis surface/bridges in the germline (PMID: 26252513). Despite the penetrant effect of this mutant on cortical remodeling during embryonic cytokinesis, both worm growth and the germlines in L1-upshifted *cyk-4(or749ts)* animals appear normal at the L4 stage; if these worms are shifted back to the permissive temperature at the L4 stage, they lay a comparable number of embryos to controls (Figure 2A in the revision). To assess the effect of the *cyk-4(or749ts)* mutation the adult, we performed the converse experiment growing the worms to the L4 stage at the permissive temperature and then shifting them to the non-permissive temperature. Under this regime, germline structure is strongly perturbed compared to equivalently treated control worms. All of the partitions in the loop and oocyte cellularization regions are absent and the proximal region of the germline has the appearance of a hollow multinucleated tube (Figure 2B in the revision). These results indicate that the Cterminal C1-GAP module of CYK-4 is important for germline morphology and function after the transition to oocyte production in the adult but is not required for worm or germline development between the L1 and L4 larval stages.

We conducted a comparable analysis for the two fast-acting temperature-sensitive mutants that disrupt the interaction between the CYK-4 and ZEN-4 dimers (the *cyk-4(t1689ts*) and *zen-4(or153ts*) centralspindlin assembly mutants). As we described in our original manuscript, if these mutants are upshifted at the L4 stage, germline structure appears normal and the worms produce embryos (presented in Figure 3E, F in the revision). Interestingly, upshifting the centralspindlin assembly mutants between the L1 and L4 stage leads to a global growth defect that results in worms that are significantly smaller than control or *cyk-4(or749ts)* worms subjected to the same protocol (now shown in new Figure 3—figure supplement 1A). Even after downshifting temperature for 24 hours, these worms remained significantly smaller than comparably treated control or *cyk-4(or749ts)* worms. The germlines in these centralspindlin assembly mutant worms, upshifted between the L1 and L4 stages, were smaller than those in control and *cyk-4(or749ts)* worms, and exhibited apparent “pathfinding” defects (new Figure 3—figure supplement 1B). The aberrant germlines were also not able to produce embryos following temperature downshift (new Figure 3—figure supplement 1C).The global effect on worm growth makes it difficult to interpret germline phenotypes because it is likely that the observed germline defects arise as a secondary consequence of the growth defect. We note that, despite the global inhibition of worm growth, the germlines exhibited compartment proliferation, and multinucleated compartments were not observed, consistent with the idea that an intact centralspindlin complex is not required for compartment proliferation per se. We note that the observation that centralspindlin assembly, but not the C1-GAP module of CYK-4, is important for growth is an unexpected result that will require future work to establish precisely where and how centraslpindlin is acting to promote growth.

In summary, our results indicate that the interaction between CYK-4 and ZEN-4, which is required for midzone assembly during cytokinesis (PMID:10871280; PMID: 11782313; PMID: 11050384), is essential for worm growth/development between the L1 and L4 stages, whereas the C-terminal C1-GAP module of CYK-4 is not. Conversely, the C-terminal CYK-4 C1-GAP module, which is required for contractile ring assembly during cytokinesis (PMID: 19056985; PMID: 22226748; PMID: 25073157; PMID: 26252513), is essential for oocyte production in adult worms, whereas ZEN-4 is not.

2) The authors argue that ZEN-4 is not required for compartment bridge closure in the germline because they did not observe any microtubule bundles passing through the compartment bridges in the pachytene, loop, or oocyte cellularization regions of the germline. However, I do see some filament-like structures in their figures, despite the low resolution. Moreover, a previous study (Wolke et al., 2007) has shown with an α-tubulin staining the presence of microtubules running from the rachis into the growing oocytes.

As the reviewer suggests, microtubules are abundant in the germline and do indeed extend into compartments and oocytes as has been demonstrated in previous papers examining tubulin in live and fixed samples (PMID: 23370148, PMID: 17507392, PMID: 26371552). It was not our intention to imply that microtubules are absent in the rachis bridges. Instead, we were highlighting that fact that organized antiparallel microtubule bundles, akin to the cytokinesis midzone, are not observed running through the rachis bridges. To demonstrate this more clearly in the revision, we have now performed a parallel analysis of embryos depleted of the microtubule bundling protein SPD-1, which is essential for assembly of the anti-parallel microtubule bundles in the midzone array during cytokinesis (PMID: 15458647). We now show that this perturbation does not alter the microtubule organization or germline architecture in the cellularization region of the germline (new panels added to Figure 3G), suggesting that our conclusion that midzone-like microtubules do not pass through the rachis bridges is correct.

Reviewer #2:In this manuscript, the authors examined the role of centralspindlin in the intercellular bridges of the C. elegans syncytial germ line. Although both subunits of centralspindlin localize to the intercellular bridges between germ cells and the rachis (or rachis surface more broadly?), ZEN-4 was dispensable for oocyte formation and only CYK-4 was required. By comparing the phenotypes of various cyk-4 and zen-4 mutants defective for the interaction between them and for the functions of CYK-4 C1 and GAP domains, they concluded that the C1 and GAP domains are important for oocyte cellularization, CYK-4 recruitment to the rachis surface and enrichment of active RhoA. Most of the experiments were carefully designed and performed. However, it remains unclear how CYK-4 contributes to the maintenance of germline structure and cellularization of oocytes as detailed below. In addition, some of the summaries of previous works are imprecise.1) Localization of CYK-4Although the localization pattern of CYK-4 in the gonad is described "to bridges" in the Abstract, "rachis surface" is used in most places in the main text. Actually, CYK-4::mNeonGreen seems to be enriched at the peripheries of the ring-like openings (Figure 1B and Figure 1—figure supplement 3). Discrimination between flat localization on the rachis surface (plasma membrane or cell cortex) and accumulation at the edges of the ring-like openings is important. For example, in the images presented in Figure 2D, accumulation at the ring-like openings seems to be weaker in zen-4(RNAi) than in the control. When CYK-4 localization is examined, images comparable to those in Figure 1—figure supplement 3 should be presented.

We now present our data analyzing CYK-4 localization following ZEN-4 depletion in the format requested by the Reviewer (included as new Figure 3D). As we describe in the first section, we think that the rachis itself is a modified intercellular bridge. Hence, the rachis bridges are essentially bridges from the cellular compartments to a bridge. We think that this is why the same set of components accumulate on the surface of the rachis and on the surface of the rachis bridges, and why both the diameter of the rachis and the diameter of the rachis bridges decrease in parallel as oocytes cellularize. It is hard to know whether components are actually more concentrated on the surface of the rachis bridges than they are on the surface of the rachis itself, or whether this is a geometric artifact (i.e. there appears to be more signal around the periphery of the rachis bridges because you are looking down the barrel of the rachis bridges – essentially projecting the intensity from a short tube onto a plane). To avoid confusion, we now describe the components as localizing to the “rachis surface and rachis bridges” or “rachis surface/bridges” throughout the text. This localization does not dramatically change in the *zen-4(RNAi)* condition.

2) Compartmentalization and incomplete cytokinesisIn contrast to the formation of the intercellular connection between the primordial germ cells (Z2 and Z3), it is not clear whether and how compartmentalization of the newly generated gem cells is coupled with mitosis. Is the compartmentalization at the distal tip of adult gonad also incomplete cytokinesis?

Yes, the compartmentalization at the distal tip of the adult gonad is also incomplete cytokinesis. As we now clarify in the revision, there are two distinct cytokinesis-like processes that contribute to germline development: (1) the incomplete cytokinesis-like events that generate the compartments that remain attached via open bridges to the rachis during development and in the mitotic region at the distal tip of the adult germline (compartment proliferation), and (2) the complete cytokinesis-like events during which the rachis bridges close to cellularize the oocytes and separate them from the germline syncytium in the adult (oocyte cellularization). Our results suggest that CYK-4 is required for oocyte cellularization but not compartment proliferation (for a detailed summary of our mutant analysis and conclusions see response to reviewer 1 (point 5) above. The proliferation of compartments during development and in the distal tip appear to occur via a very similar incomplete cytokinesislike process in which the cup-like structure that houses the nucleus is partitioned in an anaphase-coupled fashion. The complex geometry in the 3-dimensional tissue and the depth of tissue makes live imaging of this event in adults challenging, but not impossible; please refer to Figure 2 of Rehain-Bell et al. 2017, PMID: 28065311, where they have imaged this event from a top view. In these images you can see the formation of a new partition between the separated chromosomes. During this process the new partition bisects the bridge that connected the original compartment to the rachis so that both of the new compartments retain a rachis bridge.

3) Point of CYK-4 function during oogenesisAlthough the authors emphasize the role of CYK-4 in the intercellular bridge closure, this is not very strongly supported by the data presented. After compartmentalization at the distal tip, the openings of the bridge have to be stably maintained for a long time until the nucleus reaches the proximal tip and completes cellularization. Although the authors performed longitudinal live observation of the temperature-sensitive cyk-4 mutants in Figure 3, interpretation of the result is not straight-forward. First, even at time 0 of temperature upshift, the gonad of the cyk-4(or769) mutant looks different from that of the wild-type. Widening of the bridge openings is observed at the transition zone between pachytene and loop regions. In the mutant adult gonad in Figure 1D, multinucleation is observed in the distal cells. These suggest that CYK-4 plays a role in maintenance of the circular openings of the compartment bridges. To clearly discriminate oocyte cellularization from the maintenance of the bridges, normal maintenance of the bridge diameter until the moment of cellularization need to be confirmed by more detailed analyses such as Figure 1—figure supplement 3.

Our data indicates that *cyk-4* inhibition does not cause a general defect in rachis bridge stability. Our strongest evidence for this is that after *cyk-4(or749ts)* worms are upshifted for 24 hours from the L1 stage they have a normal germline structure without any multinucleate compartments, which would be expected if rachis bridges were destabilized. If the worms are returned to the permissive temperature at the L4 stage, they also have a normal broodsize (Figure 2A in the revision). Germline structure is only disrupted if the worms are held at the restrictive temperature after the point when oocyte production begins (Figure 2B in the revision). Our longitudinal timelapse analysis further shows that the disruption of germline structure initiates at the proximal end of the germline, when oocytes fail to properly bud off the rachis during oocyte cellularization, and then propagates backwards (Figure 4 in the revision).

These observations do not rule out the possibility that the rachis bridges are wider in the upshifted mutant than in the controls as the reviewer suggests. To determine if this is the case, we performed a new experiment (included as new Figure 4—figure supplement 2) comparing the width of the rachis bridges in the pachytene region in wild-type and *cyk-4(or749ts)* mutant worms held at the permissive or non-permissive temperature for 5-6 hours (timepoint chosen based on our longitudinal temporal analysis now shown in Figure 4). We used the distal-most apoptotic cell (which typically marks the point at which compartment expansion begins) as a reference point and scored the width of the rachis bridges for the compartments in the pachytene region distal to this point, measuring the size of the opening (based on the plasma membrane signal) at its widest point in the z-stack. This analysis revealed that the width of the rachis bridges in the control and *cyk-4(or749ts)* worms are comparable at the permissive temperature (2.5–3 µm; Figure 4—figure supplement 2, compare black and dark green). The width of the rachis bridges increases in both control and *cyk-*4(*or749ts)* worms after a 5.5 to 6-hour upshift to 25°C. As the reviewer suspects, the increase is somewhat larger in *cyk-*4(*or749ts)* worms than in controls (compare grey and light green). What this means is less clear. The increase could result from a mild reduction in rachis contractility; alternatively, it could be an indirect consequence of the disruption of oocyte cellularization, which could change the balance of forces in the system due to frustrated cytoplasmic flow.

4) Mechanism of CYK-4 recruitment to the rachis surfaceThe model in Figure 6 implies the role of interaction with RHO-1 for the recruitment of CYK-4 to the rachis surface. Moreover, a positive feedback loop is mentioned (subsection “Active RhoA localizes to the rachis surface and collaborates with CYK-4 to maintain germline structure”, last paragraph). However, the effect of RHO-1 depletion on CYK-4 localization was not shown. Although the authors claim that penetrant rho-1(RNAi) disrupts the rachis surface, partial rho-1(RNAi) equivalent to Figure 5D should be possible. Alternatively, acute inhibition of RHO-1 can be achieved by injection of C3 enzyme?

See response to reviewer #1, points 1 and 2.

It is counter-intuitive that a GAP promotes GTP-form of RHO-1. However, activation of RHO-1 by CYK-4 through positive regulation of ECT-2 GEF has been reported in the context of embryonic cytokinesis (Zhang and Glotzer 2015). Does ECT-2 play a role also in the gonadal function of CYK-4?

See response to reviewer #1, points 3 and 4.

5) Functionality the FP-tagged cyk-4 transgenesAlthough this is not crucial for the authors' conclusion, both the cyk-4::mNG and cyk-4::mCh strains showed 6 and 11% embryonic lethality after cyk-4(RNAi) treatment. Is this because these transgenes are not 100% functional or because RNAi-resistance is not perfect? Or, does dsRNA injection affect the quality of the embryos? Has a transgene without FP-tagging been tried?

Yes, our experiments analyzing the functions of the C1 and GAP domains (Figure 5A-C in the revision) were performed using untagged *cyk-4* transgenes that have the same re-encoded region as the mCherry tagged transgenes that we used to analyze localization. Wild-type CYK-4 expressed from the untagged transgene rescues the RNAi inhibition perfectly (100% embryonic viability; now included in new panel added as Figure 5—figure supplement 1C). This indicates that the low level of embryonic lethality (~6%; now in Figure 5— figure supplement 1D) observed when endogenous CYK-4 is depleted in the presence of the mCherry tagged *cyk-4* transgene is due to the presence of the mCherry tag.

6) Worm trap microfluidic chipThe shapes of the devices in Figure 3 and Figure 3—figure supplement 1 look different. The outline of the device should be illustrated in the supplement.

The worm trap schematics now in Figure 4 and Figure 4—figure supplement 1 have been adjusted, as suggested by the reviewer.

7) Introduction, fourth paragraphAlthough Plk1-phosphorylation sites reside in the N-terminal half of human Cyk4/MgcRacGAP1, they are not at the N-terminus of Cyk4. They are in the linker between the coiled coil and the C1 domain.

We corrected this line to read:

“Consistent with a role in activating RhoA, a region in the N-terminus of human Cyk4, between the coiled-coil and C1 domains, has been shown to be phosphorylated by Plk1 to generate a binding site for the RhoA GEF Ect2 (Burkard et al., 2009; Wolfe et al., 2009).”

8) "During this transition, centralspindlin is released from microtubules and concentrates in the midbody ring (Elia, Sougrat, Spurlin, Hurley, and Lippincott-Schwartz, 2011; Green et al., 2013; Hu, Coughlin, and Mitchison, 2012)".This is incorrect. GFP-MKLP1 appears as a disk at the midbodies (Elia 2011 and Hu 2012). The ring-like appearance of centralspindlin by immunofluorescence is known to be an artifact due to the poor accessibility of the antibodies into the central core of the midbody matrix. Although Hu 2012 described the IF image of MKLP1 as "centralspindlin remained at the center and appeared as a single ring in some images" (Hu 2012, Figure 1C), side by side comparison of GFP fusion and IF staining clearly demonstrated that GFP-MKLP1 filled the dark zone detected by IF (Hu 2012, Figure 2D). Furthermore, based on the super-resolution observation of GFP-MKLP1 by structured illumination, Elia 2011 explicitly describes it as "disk-like distribution". No single image of a clear ring-like midbody localization of centralspindlin is found in Green 2013. Finally, immuno-EM with an anti-MKLP1 antibody also supports the disk-like localization (Elad et al., (2010). Microtubule organization in the final stages of cytokinesis as revealed by cryo-electron tomography. J Cell Sci 124, 207-215). While it is possible to speculate that there might be a short time window in which centralspindlin is relocalized from a disk to a ring, no convincing data such as time-lapse super-resolution observation have been published.

Our view on this controversial issue is influenced by fluorescence microscopy data from our lab along with tomographic electron microscopy reconstructions of the intercellular bridge generated of high pressure frozen samples by the Mueller-Reichert group. The Mueller-Reichert group films embryos prior to abrupt freezing, allowing assessment of ultrastructure at precisely defined time points. Both of our groups found that microtubules are lost from the midbody about 300s prior to abscission (about 400s after furrow initiation). After this point, no microtubules are observed in the intercellular bridge by EM.

Despite the absence of microtubules, ZEN-4 strongly accumulates in the intercellular bridge over the 300s prior to abscission (400-700s post furrow initiation; please refer to Figure 1A from Konig et al., PMID 28325808, and Figure 3C from Green et al., PMID 24217623). In embryos in which assembly of the central spindle is prevented by depletion of the PRC1 homolog SPD-1, ZEN-4 is not present on midbody microtubules as the intercellular bridge forms. Yet, even in PRC1^SPD-1^ depleted embryos, ZEN-4 still accumulates at the intercellular bridge during the 300s interval prior to abscission. Thus, we are confident that centralspindlin can localize to the intercellular bridge in a microtubule-independent fashion during abscission, which we think may be analogous to its ability to target to the rachis surface/bridges in the germline.

To better describe the available data on this point, which as the reviewer correctly highlights do not depend on super-resolution data showing that ZEN-4 is re-localized from a disk to a ring, we have rewritten the text section in the Introduction as follows: “In *C. elegans* embryos, examination of the intercellular bridge by 3D electron tomography at specific timepoints following the onset of furrowing has revealed that midbody microtubules are lost ~300s prior to abscission (Konig et al., 2017). […] These results suggest that ZEN4 can target to the intercellular bridge independently of midbody microtubules, which is reminiscent of work in cultured human cells describing localization of centralspindlin to a ring-like structure during the late stages of cytokinesis (Elia et al., 2011; Hu et al., 2012).”

[Editors' note: the author responses to the re-review follow.]

The manuscript has been improved but there are some remaining issues that need to be addressed before acceptance, as outlined below:There is one piece of data that both referees would like included, since this was a question that arose during the original reviews and has been raised again. This has to do with the how cyk-4 activates rho-1 and what role does ect-2 play in this, and seems like you already have this piece of data. My suggestion would be to include data that suggests that the mechanism is different from the Glotzer lab mechanism. You can then publish the exact mechanism in a later study.This is the point that requires some experimental data.3) How does CYK-4, which is a RhoGAP, contribute toward the activation of RHO-1? If the authors envision a mechanism analogous to what the Glotzer lab showed for cytokinesis, then defects resulting from loss of CYK-4 should be rescued by activating mutations in ECT-2 or depletion of RGA-3/4."In response to point 3, raised in the original manuscript, the authors provided in their reply to reviewers a detailed explanation for the mechanism they propose for RHO-1 activation by CYK-4, which, importantly, is distinct from the mechanism proposed by the Glotzer lab. The evidence for their proposed mechanism is "In ongoing work from our group that will be the basis for another publication". Importantly, they did not add to the manuscript any new data regarding this point. Only in the Discussion they added the following sentence: "We suspect that the role of CYK-4 in generating active RhoA is mediated by the region between its coiled-coil and C1 domains that work in human cells has previously implicated in binding to the Ect2 GEF (Burkard et al., 2009; Wolfe et al., 2009)."

First, we would like to point out that the reviewer’s assertion that we “did not add to the manuscript any new data regarding this point” is not correct. In fact, we performed the exact experiment that the reviewers requested. Perhaps the reviewer/Editor missed it, as the new data was presented in the last panel of our last figure. However, in our revision, we tested whether the defects resulting from mutation of the CYK-4 GAP domain could be rescued by RGA-3/4 depletion and we included the results as a new panel in Figure 7C. The ability of the R459A Rho GTPase interface mutant to rescue the mutant phenotype hinges on whether it retains some ability to target to the cortex/membrane and promote RhoA activation that can be rendered sufficient by removal of the main negative regulators that suppress RhoA activation (RGA-3/4). Our analysis revealed that RGA-3/4 inhibition did not rescue the germline defects resulting from mutating the Rho GTPase binding interface of the GAP domain. This finding is consistent with our observation that the R459A mutant exhibits a severe localization defect in the germline (Figure 5D). As the R459A mutant has been reported to be suppressed by RGA-3/4 inhibition during cytokinesis (PMID:26252513), where ZEN-4 is essential, we speculate that the R459A CYK-4 mutant complexed with ZEN-4 may retain sufficient localization to provide partial function during cytokinesis.

In addition to the above experimentally-addressed issue, the broader questions that the reviewer is asking are: (1) how does our work in the germline relate to the prior work during cytokinesis by the Glotzer lab, and (2) how does CYK-4, which is a RhoGAP, contribute to RhoA activation.

It is important to begin this discussion by pointing out that while the work from the Glotzer lab (PMID: 26252513) showed that the Rho GTPase-binding interface of CYK-4’s GAP domain contributes to RhoA activation during embryonic cytokinesis, it did not provide evidence for a particular mechanism. As background, there is evidence from multiple systems, including the *C. elegans* embryo, that CYK-4 contributes to RhoA activation during cytokinesis. Our manuscript adds to this by showing that CYK-4 also promotes RhoA activation in the germline. A major question in the field is how the CYK-4 GAP domain contributes to this role of CYK-4. Our data in the germline suggests that the CYK-4 GAP domain is required to localize CYK-4 to intercellular bridges. Thus, CYK4 with mutations in the Rho GTPase-binding interface of its GAP domain cannot activate RhoA in the germline because the protein does not localize. In the Glotzer paper, mutations in the Rho GTPase-binding interface of the GAP domain also exhibited phenotypes consistent with reduced RhoA activation during embryonic cytokinesis. In a schematic model they proposed a new idea for why mutations in the Rho GTPase-binding interface of the CYK-4 GAP domain compromise RhoA activation: they suggested that the CYK-4 GAP domain interacts with the GEF domain of ECT-2, in a RhoA-dependent fashion, to activate it (PMID: 26252513). However, it is important to note that there was no experimental evidence that the CYK-4 GAP domain activates the ECT-2 GEF and that their model is entirely speculative. In fact, in the paper they stated that “We assayed for activation of the ECT-2 GEF activity by the CYK-4 GAP domain in vitro. However, we have not yet been able to detect stimulation of GEF activity (data not shown).” Thus, there is no evidence that the GAP domain of CYK-4 directly activates the ECT-2 GEF.

To summarize, the Rho GTPase-binding interface of the CYK-4 GAP domain contributes to RhoA activation during cytokinesis (Glotzer paper) and in the germline (this manuscript). The Glotzer paper proposes a speculative mechanism for RhoA activation, involving CYK-4 GAP domain and ECT-2 GEF domain association. Our work suggests that the CYK-4 GAP domain is required for cortical targeting of CYK-4, which is in turn required for RhoA activation. The importance of the Rho GTPase-binding interface of the GAP domain in targeting CYK-4 in the germline that we describe explains the inability of mutants in this interface to activate RhoA. Thus, we believe our work provides an explanation for the requirement of the CYK-4 GAP domain for RhoA activation in the germline. Given that the model that GAP domain directly activates the ECT-2 GEF domain lacks any experimental support, we do not believe that it is possible to experimentally address this speculative model.

To clarify these points in the revision, we have added a new section to the Discussion:

“Our analysis revealed that GFP::RGA-3, which reads out a population of active RhoA, is lost on the rachis surface and bridges following CYK-4 depletion, suggesting that CYK-4 promotes RhoA activation in the germline. […] Additional work will be needed to determine if the C1-GAP region of CYK-4 contributes to targeting CYK-4 to the cortex during cytokinesis as it does in the germline, or if it promotes RhoA activation via a distinct mechanism in this context.”

Other points that I would like you to consider and provide explanations and/or alter text accordingly.1) Possible role of CYK-4 in the establishment/maintenance of the channels that are narrow enoughI appreciate that the authors examined the influence of the cyk-4 GAP mutant on the diameter of the rachis bridges (Figure 4—figure supplement 2). Their data indicate that the bridges are open more widely in the cyk-4 mutant. This raises a possibility that the failure of cellularization could be an indirect consequence of a failure in establishment or maintenance of the channels to be narrow enough for successful cellularization. For example, one can imagine a scenario that cellarization itself is achieved without CYK-4 but can't work if the opening is too wide.

To address this point we have revised the text describing this experiment to read as follows:

“We also analyzed the width of rachis bridges in the pachytene region after 5-6 hours at non-permissive temperature (Figure 4—figure supplement 2) and found that the width of the rachis bridges increased in both control and *cyk-4(or749ts)* worms after a 5.5 to 6-hour upshift to 25°C. […] Although it is possible that the failure of oocyte cellularization in the upshifted mutants results from the fact that bridge width in the pachytene region is too large, we think this is unlikely, as rachis bridges in wild-type germlines are often observed to expand beyond a diameter of 4.5 µm as they are loaded with components in the turn region (Figure 1C).”

2) Ring-like localization of centralspindlin in mammalian cellsThe point in the previous comment was that the ring-like localization of centralspindlin in mammalian cultured cells has not been established. The arguments in the rebuttal are completely out of focus. In Hu 2012, the ring-like pattern was shown only by IF but not by live GFP-MKLP1 imaging. In Elia 2011, which employed super-resolution microscopy, the localization of MKLP1 was explicitly described as "disk-like". Immuno-EM in Elad 2010, which has been neglected by the authors, confirmed uniform localization of MKLP1 epitopes across the midbody. In their rebuttal, the authors didn't argue against these points. Nonetheless, in the current manuscript they still describe "These results suggest that ZEN-4 can target to the intercellular bridge independently of midbody microtubules, which is reminiscent of work in cultured human cells describing localization of centralspindlin to a ring-like structure during the late stages of cytokinesis (Elia et al., 2011; Hu et al., 2012)." The clause, "which is reminiscent…" should be removed since this is imprecise and misleading.

We have removed the offending clause from the Introduction. This section now reads as follows:

“In *C. elegans* embryos, examination of the intercellular bridge by 3D electron tomography at specific timepoints following the onset of furrowing has revealed that midbody microtubules are lost ~300s prior to abscission (Konig et al., 2017). […] These results suggest that centralspindlin can target to the intercellular bridge independently of midbody microtubules during abscission in *C. elegans*, perhaps consistent with the localization of centralspindlin to stable intercellular bridges that lack intervening midbodies in the syncytial germlines of different species (Carmena et al., 1998; Greenbaum et al., 2009; Greenbaum et al., 2007; Greenbaum et al., 2006; Haglund et al., 2010; Minestrini et al., 2002; Zhou et al., 2013).”